# β-arrestin1/YAP/mutant p53 complexes orchestrate the endothelin A receptor signaling in high-grade serous ovarian cancer

Piera Tocci [1], Roberta Cianfrocca[1], Valeriana Di Castro[1], Laura Rosanò[1], Andrea Sacconi[2], Sara Donzelli[2], Silvia Bonfiglio[3], Gabriele Bucci[3], Enrico Vizza[4], Gabriella Ferrandina[5], Giovanni Scambia[5], Giovanni Tonon [3,6], Giovanni Blandino [2] & Anna Bagnato [1]

The limited clinical response observed in high-grade serous ovarian cancer (HG-SOC) with high frequency of TP53 mutations (mutp53) might be related to mutp53-driven oncogenic pathway network. Here we show that β-arrestin1 (β-arr1), interacts with YAP, triggering its cytoplasmic-nuclear shuttling. This interaction allows β-arr1 to recruit mutp53 to the YAP-TEAD transcriptional complex upon activation of endothelin-1 receptors (ET-1R) in patient-derived HG-SOC cells and in cell lines bearing mutp53. In parallel, β-arr1 mediates the ET-1R-induced Trio/RhoA-dependent YAP nuclear accumulation. In the nucleus, ET-1 through β-arr1 orchestrates the tethering of YAP and mutp53 to YAP/mutp53 target gene promoters, including EDN1 that ensures persistent signals. Treatment of patient-derived xenografts reveals synergistic antitumoral and antimetastatic effects of the dual ET-1R antagonist macitentan in combination with cisplatinum, shutting-down the β-arr1-mediated YAP/mutp53 transcriptional programme. Furthermore, $ET_AR$/β-arr1/YAP gene signature correlates with a worst prognosis in HG-SOC. These findings support effective combinatorial treatment for repurposing the ET-1R antagonists in HG-SOC.

[1] Preclinical Models and New Therapeutic Agents Unit, Istituto di Ricovero e Cura a Carattere Scientifico (IRCCS), Regina Elena National Cancer Institute, 00144 Rome, Italy. [2] Oncogenomic and Epigenetic Unit, IRCCS, Regina Elena National Cancer Institute, 00144 Rome, Italy. [3] Center for Translational Genomics and Bioinformatics, IRCCS, San Raffaele Scientific Institute, 20132 Milan, Italy. [4] Gynecologic Oncology, IRCCS, Regina Elena National Cancer Institute, 00144 Rome, Italy. [5] Gynecologic Oncology, Fondazione Policlinico Universitario A. Gemelli, IRCCS, Catholic University of Rome, 00168 Rome, Italy. [6] Functional Genomics of Cancer Unit, Division of Experimental Oncology, IRCCS, San Raffaele Scientific Institute, 20132 Milan, Italy. Correspondence and requests for materials should be addressed to G.B. (email: giovanni.blandino@ifo.gov.it) or to A.B. (email: annateresa.bagnato@ifo.gov.it)

High-grade serous ovarian cancer (HG-SOC), accounts for 70–80% of ovarian cancer deaths and is characterized by the highest rate of recurrence and TP53 mutations[1]. Standard treatment for HG-SOC is surgical debulking followed by platinum-based chemotherapy. While chemotherapy is initially effective, the majority of HG-SOC patients succumbs to recurrent and chemoresistant disease. Thus, there is an unmet medical need to develop more effective therapeutic approaches for HG-SOC[2]. The appropriate control of drug response requires the integration of numerous signaling pathways operating concomitantly within tumor cells. Therefore, effective treatments would require a comprehensive understanding of the nodes and networks underlying drug response and metastatic progression. The endothelin-1 (ET-1) receptors (ET-1R) $ET_AR$ and $ET_BR$, members of G-protein coupled receptors (GPCR) family, are well-known drivers of tumor progression in many human malignancies, including OC[3]. Of clinical relevance, the association between $ET_AR$ expression and poor survival is to be ascribed to the unfavorable prognostic role of $ET_AR$ in the subset of platinum-resistant OC cases[4]. In the canonical GPCR signaling agenda, receptor activation results in the stimulation of heterotrimeric G proteins to initiate intracellular signaling pathways. Besides the crucial role of G-protein-dependent signaling in cancer, recent studies have revealed the key contribution of β-arrestin1 (β-arr1)-mediated signaling to GPCR function in malignant diseases[5–7]. In OC cells, β-arr1 has emerged as dynamic multitask scaffold protein, organizing complex signaling networks involved in multiple $ET_AR$-dependent mechanisms, such as cell proliferation, survival, invasion, migration, neovascularization, and chemoresistance[3,7]. In addition to β-arr1-guided control of cytosolic and cytoskeletal functions, several studies depicted a nuclear role for β-arr1[8–12]. In ET-1-stimulated OC cells, β-arr1 acts as a dynamic nuclear linker of several transcriptional factors and co-factors, such as β-catenin[13,14], NF-κB[15] and the hypoxia-inducible factor 1α (HIF-1α)[16], mediating the epigenetic regulation of genes strictly associated to OC growth and progression. Therefore, the deep knowledge of the β-arr1 interactome and how targeting the $ET_AR$/β-arr1-interconnected routs represent a substantial therapeutic challenge to improve patient survival.

Among the oncogenic drivers activated by GPCR and involved in OC progression and resistance to therapy, Yes-associated protein (YAP) and WW Domain Containing Transcription Regulator (WWTR or TAZ), two related transcriptional co-activators of Hippo pathway, have become the focus of intense research and have been implicated in worse patient prognosis[17–26]. In OC cells, YAP and TAZ confer resistance to conventional chemotherapeutics, including cisplatinum[22] and promote the formation of ascites and metastatic nodules[23]. The core components of the GPCR-driven Hippo pathway establish a kinase cascade which results in the inhibition of LATS1/2 kinase activity, followed by YAP/TAZ nuclear translocation and their binding to the TEA domain (TEAD1–4) family transcription factors, and activation of their target gene expression. It has been previously reported that in colon cancer cells $ET_AR$ triggers YAP/TAZ activation through $G\alpha_{q/11}$[27], and in OC cells lysophosphatidic acid (LPA) receptor induced $G_{12/13}$/Rho/YAP pathway[28], implying that regulation of YAP/TAZ pathway by GPCR is functionally and structurally centered on G-protein-dependent pathway. Despite these findings, there are still open questions regarding the interplay between GPCR and YAP/TAZ activity in OC, what are the molecular determinants triggering it, and what are the functional consequence of the input blockade.

Recent evidence demonstrates that the Hippo pathway cooperates with other deregulated pathways in a process that amplifies YAP/TAZ activity[29]. There are several evidences of crosstalk between mutant p53 (mutp53) protein and the Hippo pathway[30,31]. In particular, mutp53, characterizing over 90% of HG-SOC[32], and YAP and TAZ can physically associate with each other in the nucleus to drive pro-proliferative gene expression to promote tumorigenesis, suggesting that mutp53 and YAP/TAZ might represent a critical oncogenic node that shares a common transcriptional signature critical for metastatic outgrowth in different human malignancies[33–35].

Here, in patient-derived HG-SOC cells and in tumor cell lines with hot-spot missense TP53 mutations, we characterize the molecular interaction between β-arr1 and YAP that allows the formation of the nuclear complex mutp53/TEAD/YAP to coordinate transcriptional responses to ET-1. ET-1R blockade by macitentan, a dual $ET_AR$/$ET_BR$-antagonist, FDA-approved for the treatment of pulmonary arterial hypertension[3], sensitizes tumor cells to different cytotoxic and targeted agents in various preclinical tumor models, including ovarian, colorectal cancer, glioblastoma, multiple myeloma, breast, and lung brain metastasis[4,36–43], and may represent an effective therapeutic option for cancer patients. Therefore we hypothesize that macitentan can enhance the activity of platinum-based therapy in HG-SOC dampening the oncogenic interplay between ET-1R/β-arr1 and YAP/mutp53.

## Results

**ET-1/$ET_AR$ axis promotes YAP/TAZ nuclear shuttling in HG-SOC.** Taking into account that hyperactivated YAP plays a critical role in controlling the fate of OC cells[19–23] and considering the ability of ET-1/$ET_AR$ axis to promote the acquisition of plasticity, and resistance to chemotherapy, associated with OC progression[4,38,44], we evaluated the potential role of ET-1/$ET_AR$ signaling to activate YAP/TAZ pathway in OC cells. As preclinical relevant models we used ascites patient-derived, early passage primary cultures of HG-SOC cells (PMOV10), which recapitulate the HG-SOC features. PMOV10 cells were characterized for the copy number expression of ET-1, $ET_AR$ and β-arr1 (Supplementary Fig. 1a). TP53 gene sequencing of PMOV10 cells displayed a single nucleotide (C > G) germline missense mutation variant (R337T) on the exon 9 (Supplementary Fig. 1b). In parallel, we used OVCAR-3, a HG-SOC cell line that carries one of the hot spot missense TP53 mutation (R248Q). In both PMOV10 and OVCAR-3 cells, that expressed $ET_AR$ and $ET_BR$ (Supplementary Fig. 1c), ET-1, in a time-dependent manner, promoted the reduction of pYAP (S127) and pTAZ (S89) in the cytoplasm paralleled with YAP/TAZ nuclear accumulation (Fig. 1a, b). Similarly, in a breast cancer cell line (MDA-MB-468) carrying a common TP53 mutation (R273H) and expressing $ET_AR$ and $ET_BR$ (Supplementary Fig. 1c), ET-1 induced YAP/TAZ dephosphorylation and nuclear accumulation (Fig. 1c). The dual $ET_AR$/$ET_BR$ antagonist macitentan prevented YAP/TAZ nuclear accumulation (Fig. 1d, e) in both HG-SOC cells analyzed. The ET-1-dependent YAP nuclear accumulation was also confirmed by immunofluorescence (IF) analysis in OVCAR-3 (Fig. 1f), as well as in PMOV10 cells (Supplementary Fig. 1d). To rule out any off-target effects of ET-1R antagonist, we observed that the ET-1-dependent YAP nuclear accumulation was strongly inhibited upon pharmacological ET-1R blockade with macitentan, or with the $ET_AR$ selective antagonist BQ123, or with the combination of BQ123 and the $ET_BR$ selective antagonist, BQ788. Silencing of $ET_AR$ was able to recapitulate the effect of ET-1R antagonist. On the contrary, the addition of only the $ET_BR$ antagonist BQ788, as well as the depletion of $ET_BR$, did not decrease YAP nuclear accumulation, demonstrating that ET-1 acts mainly through $ET_AR$ to control the nuclear trafficking of YAP in HG-SOC cells (Supplementary Fig. 1f–h). Moreover, we

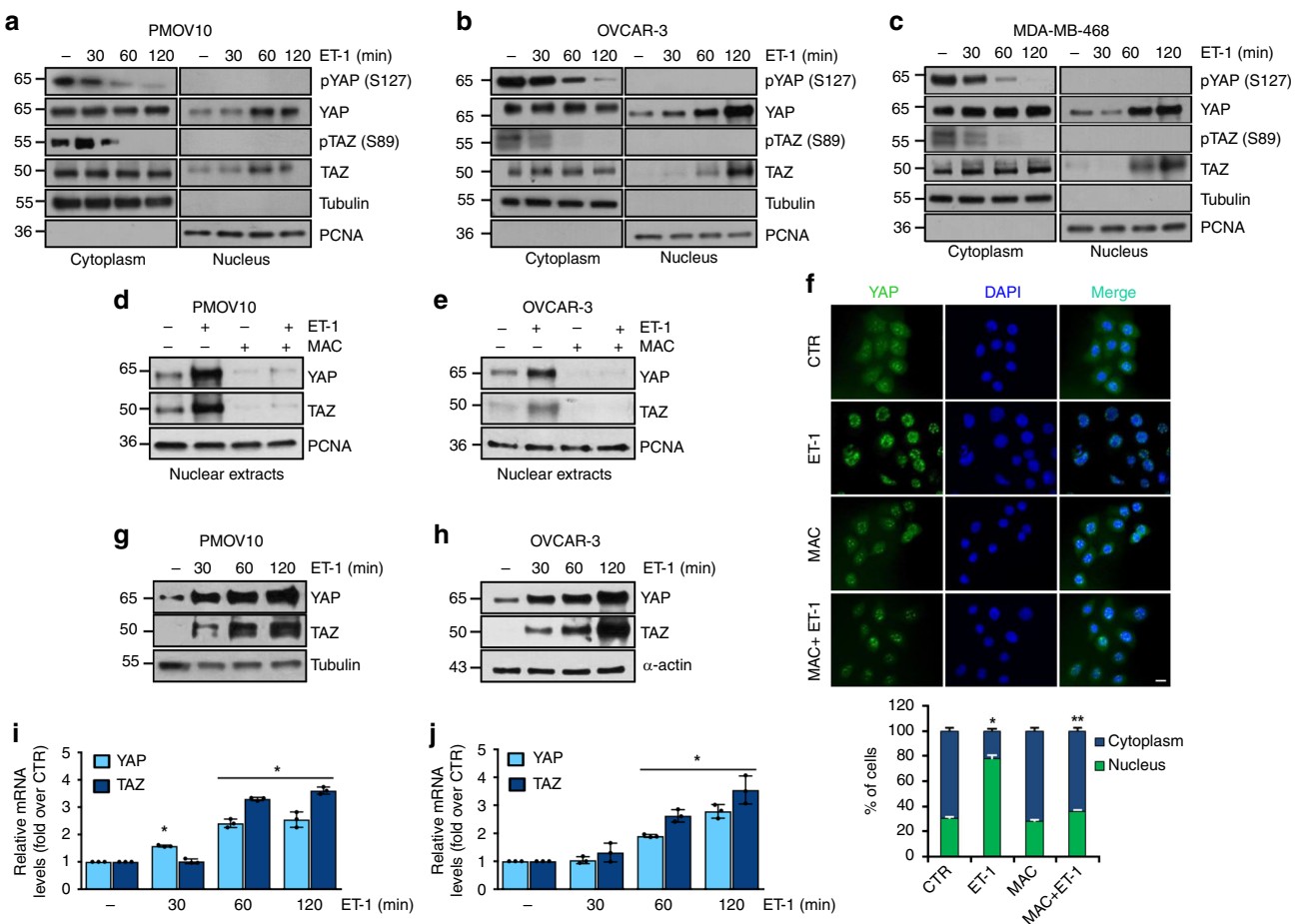

**Fig. 1** ET-1 promotes YAP/TAZ cytoplasmic-nuclear shuttling in mutant p53 tumor cells. **a–c** IB analysis of pYAP (S127), YAP, pTAZ (S89) and TAZ protein expression in the nuclear and cytoplasmic extracts of PMOV10 (**a**), OVCAR-3 (**b**), and MDA-MB-468 (**c**) cells stimulated with ET-1 (100 nM) for the indicated times. Tubulin and PCNA were used as cytoplasmic and nuclear loading control, respectively. **d, e** IB analysis of YAP and TAZ protein expression in the nuclear extracts of PMOV10 (**d**), and OVCAR-3 cells (**e**) upon stimulation with ET-1 (100 nM) and/or macitentan (MAC; 1 μM) for 90 min. PCNA was used as loading control. **f** YAP localization evaluated by immunofluorescence (IF) in OVCAR-3 stimulated with ET-1 and/or MAC for 90 min. Nuclei are stained in blue (DAPI). Bottom graph represents the quantification of YAP nuclear vs. cytoplasmic localization. Scale bar, 20 μm. Bars are means ± SD; $*p < 0.0001$ vs CTR, $**p < 0.0001$ vs ET-1 ($n = 3$). **g–j** IB and qRT-PCR analysis of YAP and TAZ protein and mRNA expression, respectively, in PMOV10 (**g, i**) and OVCAR-3 (**h, j**) cells stimulated with ET-1 (100 nM) for the indicated times. Tubulin or α-actin was used as loading control. Bars are means ± SD; $*p < 0.001$ vs CTR ($n = 3$)

explored whether ET-1 regulates total YAP/TAZ expression. As shown in the Fig. 1g–j, ET-1 upregulates the YAP and TAZ at both mRNA and protein levels, suggesting that ET-1 induces YAP/TAZ expression at the transcriptional level. Collectively, these results establish that ET-1 upregulates YAP/TAZ expression and that ET$_A$R activation by ET-1 induces YAP/TAZ de-phosphorylation and nuclear enrichment in HG-SOC cells.

**β-arr1 mediates ET-1/ET$_A$R-dependent YAP nuclear localization.** β-arr1 is characterized for its function to mediate cytoplasmic and nuclear signaling of activated GPCR with respect to heterotrimeric G protein signaling[45]. The current model for the GPCR-dependent regulation of YAP/TAZ pathway is focused on G-protein activity[24–26]. Considering the key contribution of β-arr1-mediated nuclear signaling to ET$_A$R function, in which β-arr1 establishes different transcriptional partnerships enhancing gene transcription[3,6,13–16], we evaluated whether consequently to ET-1/ET$_A$R axis activation, β-arr1 could functionally contribute to YAP activity regulation. To address this hypothesis we performed co-immunoprecipitation analysis in cytoplasmic and nuclear extracts derived from PMOV10 cells, and found that

endogenous β-arr1 physically interacted with endogenous YAP into the cytoplasm, after 30 and 60 min of ET-1 activation. This led to β-arr1/YAP nuclear co-translocation and interaction, after 90 min of ET-1 stimulation (Fig. 2a) that occurred also at longer time points as for 12/24 h (Supplementary Fig. 4b). Of note macitentan treatment impaired this interaction (Fig. 2b). To explore the nuclear function of β-arr1, we transfected PMOV10 cells stimulated with ET-1 with a construct expressing a β-arr1 deletion mutant lacking the sequence required for its nuclear localization and tagged with an AU5 epitope at its carboxyl termini (β-arr1-1-180N)[14,46]. We observed that the deletion mutant was unable to co-immunoprecipitate with YAP (Fig. 2c). β-arr1 silencing, as well as macitentan treatment, strongly reduced YAP/TAZ nuclear accumulation (Fig. 2d and Supplementary Fig. 2a, b). To further evaluate the role of β-arr1 in the cytoplasmic-nuclear shuttling, we transfected ovarian cancer cells with β-arr1Q394L mutant, in which the nuclear export signal was introduced into β-arr1, by a single point (Q394L) mutation[47–49]. The cells transfected with β-arr1Q394L, were characterized by a reduction of YAP/TAZ nuclear accumulation, which was rescued by the re-expression of β-arr1 (Fig. 2d and Supplementary Fig. 2a, b). These findings clearly indicate that β-arr1 functions as

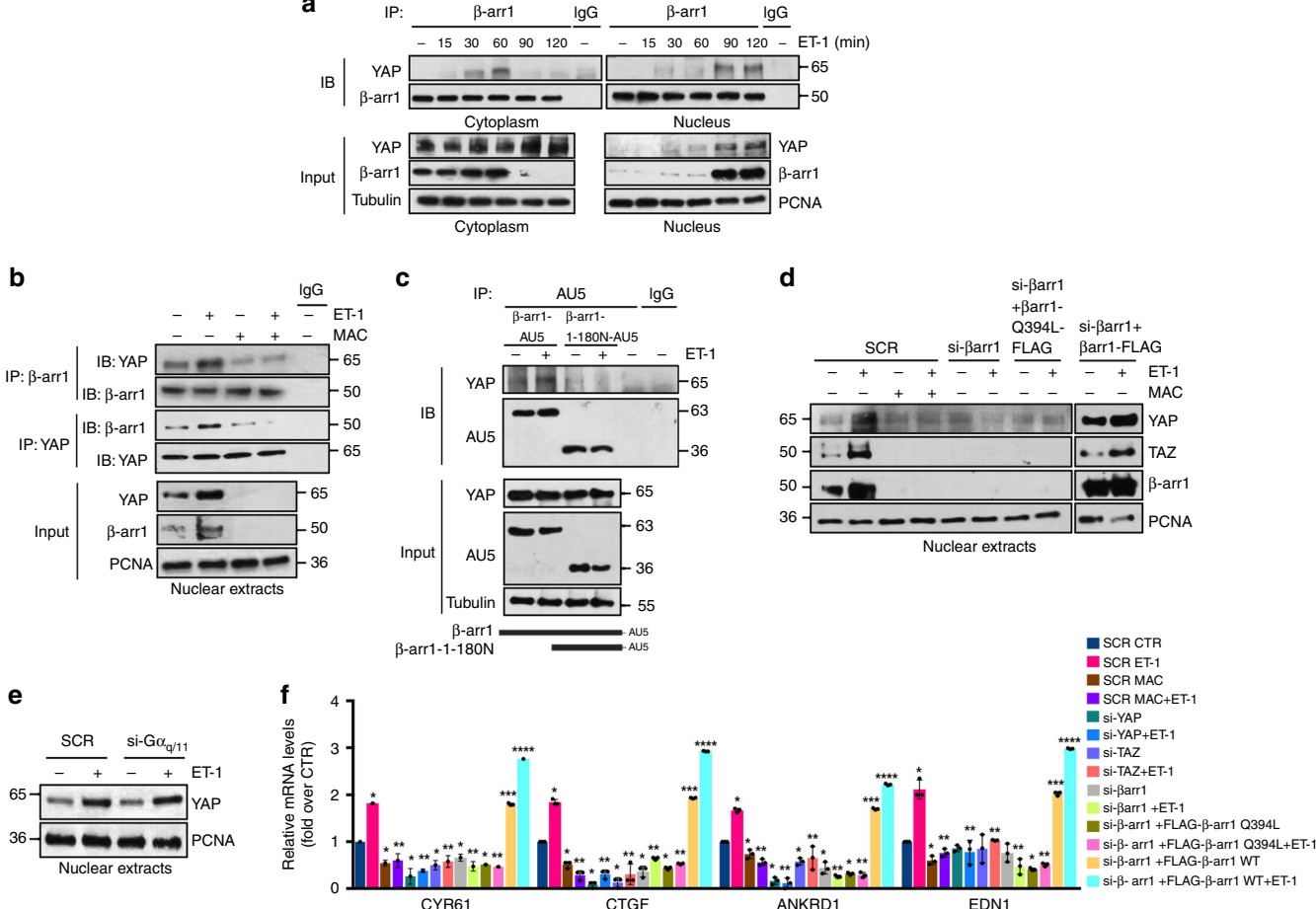

**Fig. 2** β-arr1 interacts with YAP mediating the ET-1/ET$_A$R-dependent YAP nuclear activity. **a** Cytoplasmic and nuclear extracts of PMOV10 cells treated with ET-1 for the indicated times, were immunoprecipitated (IP) for endogenous β-arr1 or for YAP, or anti-immunoglobulin G (IgG) as control Ab and IB using anti-β-arr1 and anti-YAP. Tubulin and PCNA were used as loading controls. **b** Nuclear extracts of PMOV10 cells stimulated with ET-1 for 90 min and/ or MAC were IP for endogenous β-arr1 using anti-β-arr1, or anti-IgG and IB using anti-β-arr1 and anti-YAP. PCNA was used as loading control. **c** Total extracts of PMOV10 cells treated or untreated with ET-1 for 90 min, transfected with the full-length β-arr1-AU5 and the mutant β-arr1-1-180N-AU5, were IP with anti-AU5, or anti-IgG and IB using anti-AU5 and anti-YAP. (Bottom, full-length β-arr1-AU5 and deletion mutant β-arr1-1-180N-AU5). Tubulin was used as loading control. **d** IB analysis for YAP, TAZ and β-arr1 upon stimulation with ET-1 and/or MAC for 90 min in nuclear extracts of PMOV10 cells transfected with SCR, or si-β-arr1, or si-β-arr1 and mutant β-arr1Q394L-FLAG, unable of nuclear localization, and then rescued with β-arr1-FLAG. PCNA was used as loading control. **e** IB analysis for YAP in PMOV10 cells silenced for Gα$_{q/11}$ for 72 h and treated with ET-1 for 90 min. **f** Expression analysis (qRT-PCR) of the indicated YAP target genes in PMOV10 cells stimulated with ET-1 for 24 h and treated with MAC or transfected with the indicated si-RNA or with mutant β-arr1Q394L-FLAG, and then rescued with β-arr1-FLAG. Bars are means ± SD (*$p < 0.01$ vs CTR; **$p < 0.01$ vs ET-1; ***$p < 0.03$ vs β-arr1 silenced cells; ****$p < 0.05$ vs β-arr1 silenced cells treated with ET-1) ($n = 3$)

a chaperone of ET$_A$R in regulating YAP nuclear localization. Moreover, we analyzed whether Gα$_{q/11}$, the main G protein involved, could also contribute to the ET-1R-mediated YAP cytoplasmic-nuclear shuttling. IB analysis in nuclear extracts of PMOV10 cells revealed that Gα$_{q/11}$ silencing did not affect ET-1R-induced YAP nuclear accumulation, demonstrating that β-arr1 mediates YAP nuclear shuttling in a G protein-independent manner (Fig. 2e and Supplementary Fig. 2c). Along with these results, quantitative real-time PCR (qRT-PCR) performed in PMOV10 (Fig. 2f and Supplementary Fig. 2b–d), and OVCAR-3 (Supplementary Fig. 2f–i), revealed that ET-1 stimulus induced the express ion of well-known YAP direct target gene[50] levels, such as *CYR61*, *CTGF*, *ANKRD1*, and *EDN1*. Macitentan treatment, as well as β-arr1, or YAP, or TAZ silencing, or by the ectopic expression of β-arr1Q394L in β-arr1-depleted cells, counteracted this effect. On the contrary, YAP target gene expression was fully rescued upon re-expression of β-arr1. Taken together, our results prove the critical role of β-arr1 in

transducing ET-1/ET$_A$R-dependent YAP cytoplasmic-nuclear shuttling, via a hitherto unveiled β-arr1-YAP physical interaction independent of the G-protein.

## ET-1R/β-arr1 activates YAP through LATS, Trio, RhoA, and actin.
Activating mutations of G-proteins can trigger Trio, a RhoGEF family member, leading to YAP nuclear translocation and activation[26]. Because in OC cells ET$_A$R/β-arr1 interacts with a member of RhoGEF family, the post-synaptic density protein 95/disc-large/zonula occludens-RhoGEF (PDZ-RhoGEF), to activate Rho signaling[51,52], we investigated whether β-arr1 could contribute to YAP activity by interacting with another member of RhoGEF family, as Trio. ET-1 stimulation enhanced, in a time-dependent manner, Trio detection in β-arr1 immunoprecipitates of PMOV10 cells (Fig. 3a). Moreover, ET-1 stimulation increased Trio-RhoGEF tyrosine-phosphorylation (Supplementary Fig. 3a), which commonly regulates RhoGEF activity. β-arr1 silencing

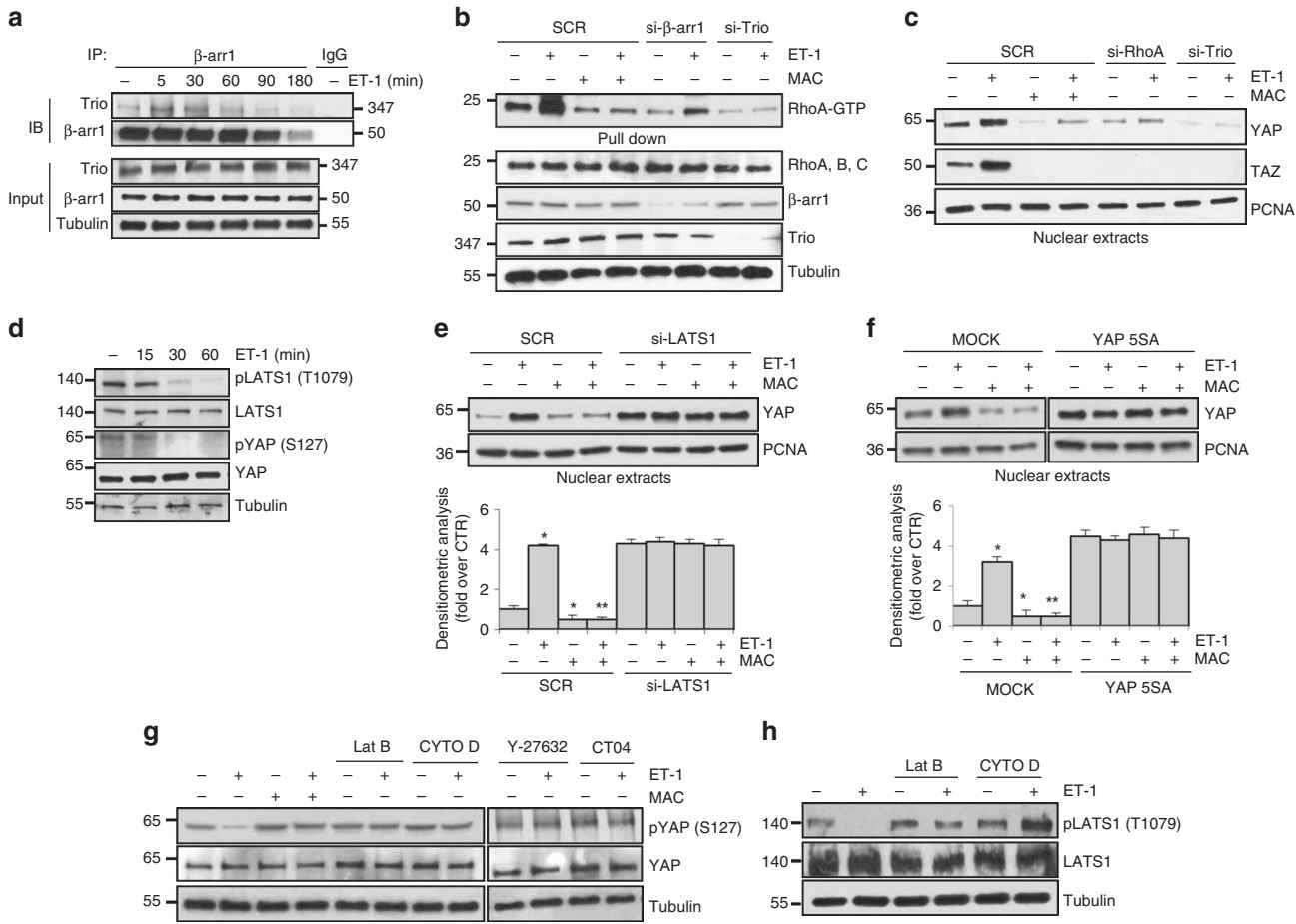

**Fig. 3** ET-1R/β-arr1 mediates YAP activity. **a** Total extracts of PMOV10 cells stimulated with ET-1 for different times were IP for endogenous β-arr1 by using anti-β-arr1 or anti-IgG and IB using anti-β-arr1 and anti-Trio. Tubulin was used as loading control. **b** Rhotekin was used to pull down RhoA-GTP from total lysates of PMOV10 cells transfected with SCR or si-β-arr1 or si-Trio and stimulated with ET-1 and/or MAC for 5 min. The GTP pull-down and input were then analyzed by IB. **c** IB analysis for YAP and TAZ expression in the nuclear extracts of PMOV10 cells transfected with SCR or si-RhoA or si-Trio, or treated with MAC, and stimulated with ET-1 for 90 min. PCNA was used as loading control. **d** PMOV10 cells stimulated with ET-1 for the indicated times were IB for pLATS1 (T1079), LATS1, pYAP (S127) and YAP. Tubulin was used as loading control. **e, f** PMOV10 cells transfected with SCR or si-LATS1 for 72 h (**e**), or with an empty vector (MOCK) or with a vector encoding for YAP constitutively active (YAP 5SA-Myc) for 24 h (**f**) and stimulated with ET-1 and/or MAC for 90 min were IB for YAP. Bottom graphs: quantification of YAP normalized to PCNA (Bars are means ± SD; *p < 0.001 vs SCR CTR; **p < 0.001 vs SCR ET-1) (**e**); *p < 0.001 vs MOCK CTR; **p < 0.001 vs MOCK ET-1) (**f**). **g** PMOV10 cells treated with ET-1 and/or MAC, were additionally treated with disruptors of actin cytoskeleton filaments, Latrunculin B (Lat B) or Cytochalasin D (CYTO D), or with a specific ROCK1 inhibitor, Y-27632, or with a specific RhoA inhibitor, CT04 and IB for pYAP (S127) and YAP. **h** PMOV10 cells treated with ET-1 were treated with Lat B or CYTO D, and IB for pLATS (T1029) and LATS1. Tubulin was used as loading control

diminished Trio-RhoGEF phosphorylation (Supplementary Fig. 3a), supporting the hypothesis that β-arr1/Trio physical interaction could modulate Trio activity. In line with these results, β-arr1 and Trio depletion, as well as macitentan treatment, decreased ET-1R-induced RhoA GTPase activity (Fig. 3b and Supplementary Fig. 3b). Intriguingly, either Trio silencing, or RhoA GTPase depletion and macitentan treatment hampered the ET-1-induced YAP/TAZ nuclear accumulation (Fig. 3c and Supplementary Fig. 3b), proving that $ET_AR$/β-arr1/Trio pathway promotes RhoA GTPase activity to regulate YAP/TAZ pathway.

To address whether LATS was involved in YAP regulation by ET-1 in OC cells, we used different approaches. First, we detected that in parallel with decreased YAP phosphorylation, ET-1 caused, in a time-dependent manner, the reduction of the phosphorylated and active form of LATS1 (pLATS1-T1079) (Fig. 3d). Moreover, LATS1 silencing strongly prevented the ET-1 and/or macitentan effects on YAP nuclear accumulation (Fig. 3e and Supplementary Fig. 3c). To further confirm the role of LATS

in ET-1-induced YAP nuclear accumulation, we transfected PMOV10 cells with a construct expressing the mutant form YAP5SA[53], resistant to LATS-mediated phosphorylation. Cells transfected with YAP5SA were almost insensitive to ET-1 and/or macitentan treatment (Fig. 3f and Supplementary Fig. 3d), suggesting that in HG-SOC cells, $ET_AR$/β-arr1 pathway, interfering with LATS kinase activity, regulates YAP nuclear accumulation.

To assess whether actin cytoskeleton remodeling was involved in ET-1-induced YAP regulation, we used different chemical inhibitors. The treatment of PMOV10 cells with actin disrupting agents, such as Latrunculin B (Lat B) and Cytochalasin D (CYTO D), as well as the use of a selective ROCK inhibitor, Y-27632, and of a RhoA GTPase inhibitor, CT04, likewise treatment with macitentan, blocked the ET-1-induced de-phosphorylation of YAP (Fig. 3g). Interestingly, actin disrupting agents also blocked the ET-1-induced de-phosphorylation of LATS (Fig.3h). Altogether, these results indicate that the regulation of LATS activity

requires a functional actin cytoskeleton and suggest that LATS contributes to ET-1R/β-arr1/Trio/RhoA-mediated activation of YAP signaling.

**β-arr1/YAP/mutp53 complex mediates ET-1R-induced transcription.** The interaction of mutp53 proteins with transcription regulators, and other cellular proteins can reshape the tumor cell transcriptome, subverting crucial pathways and fostering cell proliferation, survival, invasion, metastasis, and chemoresistance[54]. Immunohistochemistry (IHC) analysis of HG-SOC patient, from which PMOV10 cells derived, showed a strong YAP expression along with a robust p53 staining localized in the nuclear compartment (Supplementary Fig. 1i). Interestingly, mutp53 protein was further accumulated and paired with YAP nuclear accumulation upon ET-1 stimulation, (Supplementary Fig. 1d, e). Since it has been demonstrated that in different models of human cancers, mutp53 interacts with YAP[30–34], we investigated by combinatorial IP whether downstream of ET-1R/β-arr1, YAP could be a part of a functional axis that includes also mutp53 in the nucleus. Co-immunoprecipitation (co-IP) assays in PMOV10, and OVCAR-3 cells, upon ET-1 stimulus, showed that YAP and mutp53 were engaged in a trimeric complex with β-arr1, that was disrupted following macitentan treatment (Fig. 4a, b). Interestingly, we confirmed the presence of the trimeric complex β-arr1/YAP/mutp53 in MDA-MB-468 breast cancer cells carrying mutp53 R273H (Supplementary Fig. 4a). We found that β-arr1-dependent signaling can generate long-lasting effects through the formation of a multi-protein complex, that occurred also at longer time points as for 12/24 h (Supplementary Fig. 4b). Of note, mutp53 depletion hampered YAP and β-arr1 interaction (Fig. 4c, d and Supplementary Fig. 4c, d), and similarly YAP silencing impaired β-arr1 and p53 binding (Fig. 4e, f and Supplementary Fig. 2 c, h). Consistent with these results, reciprocal co-IP by using p53 immunoprecipitates, revealed that β-arr1 silencing induced a loss in the ability to bind the other two components of the complex (Fig. 4g). This suggests that β-arr1 may provide a nuclear anchor for mutp53 and YAP to activate the expression of downstream target genes. These results indicate that β-arr1-dependent signaling can engender highly characteristic transcriptomic phenotypes through the formation of the β-arr1/mutp53/YAP complex.

Although YAP binds other transcription factors, TEAD seems to mostly mediate the oncogenic functions of YAP[53]. Having unveiled a multimeric nuclear complex consisting of β-arr1, YAP and mutp53, we performed chromatin immunoprecipitation (ChIP) analysis followed by PCR (Fig. 4h–k) or qRT-PCR (Supplementary Fig. 4g–j), revealing that in PMOV10 and OVCAR-3 cells, ET-1 induced the concomitant recruitment of β-arr1, YAP, TEAD and mutp53 on YAP/TEAD-responsive target gene promoters, such as CTGF, ANKRD1 and EDN1. Either macitentan treatment or β-arr1 silencing abrogated all the promoter recruitments. Moreover, upon ET-1 stimulation, β-arr1, YAP, mutp53 and TEAD co-occupied the same promoter regions on the YAP/TEAD target gene promoters, as shown by ChIP-re-ChIP assays (Fig. 4l and Supplementary Fig. 4k.). This further confirmed that nuclear β-arr1 was involved in recruiting all the partners of this transcriptional competent complex on the target genomic regions of TEAD. Of note, in MDA-MB-468 cells, where YAP and mutp53 proteins physically interact with NF-Y on genes controlling cell proliferation[34], ET-1 induced also the concomitant recruitment of β-arr1, YAP, NF-Y and mutp53 on the promoters of cell cycle genes, cyclin A (CCNA2) and cyclin B (CDK1) (Supplementary Fig. 4l), indicating that β-arr1, acting as a tethering hub, coordinates the interactions with active transcriptional factors, as either NF-Y or TEAD, that recruits

YAP with mutp53 to the DNA and regulate transcriptionally gene expression. To further explore the functionality of this transcription competent nuclear complex, we analyzed the transcription of this set of target genes, such as CYR61, CTGF, ANKRD1 and EDN1, as well as of the pro-proliferative genes CCNA and CDK1. qRT-PCR performed in PMOV10 cells (Fig. 4m, n and Supplementary Fig. 4c, d, e and f), revealed that TEAD and mutp53 depletion strongly reduced the ET-1-induced expression of target genes. In line with these findings, the analysis of TEAD transcriptional and ET-1 promoter activities performed in PMOV10 cells, revealed that these ET-1-induced activities were hampered following macitentan treatment or upon the depletion of each component of the functional transcriptional complex (Fig. 4o, p). Of relevance, the re-expression of β-arr1, but not of β-arr1 mutant, restored both TEAD transcriptional and ET-1 promoter activities. Collectively, the above data demonstrate that ET-1R activation promotes the β-arr1/mutp53 cooperation to support the transcription of YAP-dependent signature that appears to be instigated by the functional β-arr1/mutp53 network (Fig. 4q). Remarkably, ET-1 gene transcription is regulated by YAP/mutp53/TEAD transcriptional complex along with β-arr1, suggesting that the $ET_AR$-induced feed-forward loop of ET-1 may represent a magnifying persistent mechanism in HG-SOC that sustains aggressive traits.

**ET-1R/β-arr1 axis drives invasion through mutp53/YAP complex.** It has been reported that ET-1R/β-arr1 axis supports cell invasion and progression in OC[3,6,7,51,52]. Interestingly, in PMOV10 cells we observed that the ET-1-induced matrix metalloproteinase-9 (MMP-9) activation, as determined by gelatin zymography, was inhibited by β-arr1, YAP, TAZ and mutp53 silencing, as well as by macitentan treatment (Supplementary Fig. 5a). This confirmed that YAP/TAZ are involved in the MMP-9-mediated proteolytic activity upon ET-1R/β-arr1 pathway activation. YAP, or TAZ, or β-arr1 or mutp53 depletion, as well as macitentan treatment, inhibited the ET-1-enhanced cell invasive abilities of PMOV10 cells (Supplementary Fig. 5b). Altogether, these results highlight the critical role of $ET_AR$/β-arr1 axis, mediating YAP activation, to sustain an invasive HG-SOC behavior, which can be hampered by targeting ET-1R/β-arr1 pathway using macitentan.

**Macitentan sensitizes to platinum impairing ET-1R/β-arr1/YAP axis.** ET-1/$ET_AR$ axis, through β-arr1, delivers signals implicated in the acquisition of resistance to chemotherapeutic-induced apoptosis, thereby sustaining survival pathways in OC cells[4,44]. Moreover, it has been reported that YAP confers resistance of OC cells to cisplatin and taxol, and YAP depletion promotes partial sensitivity to chemotherapeutic agents[22]. Based on these findings, to evaluate whether YAP is required for ET-1R/β-arr1-mediated cell survival in HG-SOC, we showed that in PMOV10 cells macitentan treatment, as well as YAP and β-arr1 depletion, strongly reduced ET-1-induced cell growth (Fig. 5a). Consistent with these results, we found that β-arr1 and YAP silencing, as well as macitentan treatment of PMOV10 cells increased the expression of cleaved-Poly (ADP ribose) polymerase (PARP) (Fig. 5b). Next, we evaluated the response of PMOV10 cells to different doses of cisplatinum[55], and we found that the combination of cisplatinum with macitentan, rendered PMOV10 cells more prone to cisplatinum-induced cell death (Fig. 5c). Similarly to the effect of macitentan treatment, YAP depletion reduced the number of viable PMOV10 cells (Fig. 5d). Interestingly, cells silenced for YAP and co-treated with macitentan and/or CIS, were rendered more sensitive to cisplatinum. Contextually, apoptosis, as revealed by cleaved-PARP expression

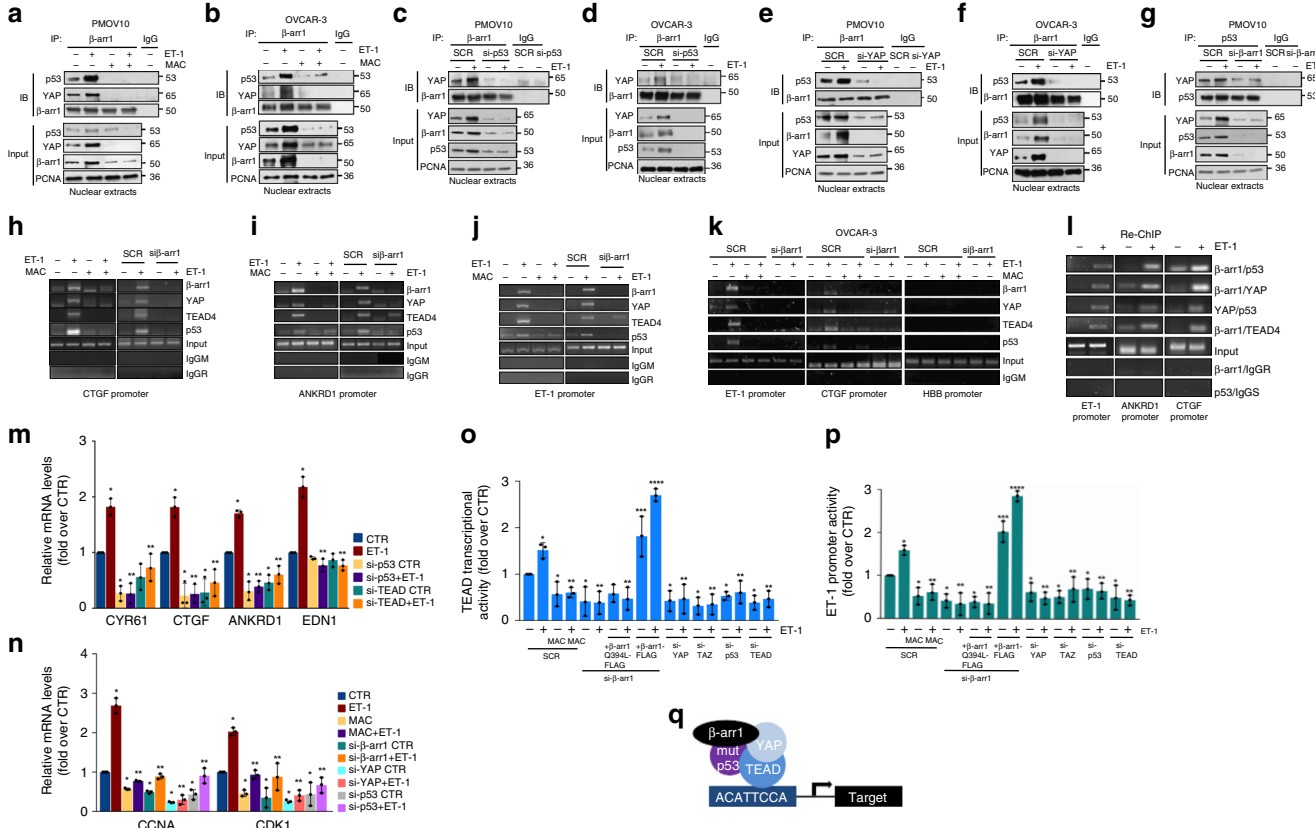

**Fig. 4** β-arr1/YAP/ mutant p53 complex mediates ET-1R-induced gene transcription. **a**–**f** PMOV10 and OVCAR-3 cells stimulated with ET-1 and/or MAC for 90 min (**a**, **b**) or silenced for p53 (**c**, **d**), or YAP (**e**, **f**) were IP for endogenous β-arr1 using anti-β-arr1, or anti-IgG Abs and IB using anti-β-arr1, anti-p53 and anti-YAP. **g** PMOV10 cells treated or no with ET-1 and silenced for β-arr1 were IP for endogenous p53 using anti-p53, or anti-IgG Abs and IB using anti-p53 and anti-YAP. PCNA was used as loading control. **h**–**k** PMOV10 (**h**–**j**) or OVCAR-3 (**k**) cells stimulated with ET-1 and/or MAC for 90 min were transfected with SCR or si-β-arr1. The binding of β-arr1, YAP, TEAD and p53 on the promoters of *CTGF* (**h**, **k**), *ANKRD1* (**i**), *ET-1* (**j**, **k**), and *HBB* (hemoglobin; negative control locus) (**k**) was measured by ChIP analysis. Anti-IgG mouse or rabbit Abs (IgGM, IgGR) were used as control for all ChIP reactions. **l** The co-occupancy of β-arr1/YAP/TEAD/p53 to *ET-1*, *ANKRD1* and *CTGF* promoter was measured in PMOV10 cells by ChIP-re-ChIP assays. Non-specific anti-IgG mouse or sheep Abs (IgGM, IgGS) were used as control for all ChIP reactions. **m**, **n** Expression analysis (qRT-PCR) of the indicated YAP/TEAD mRNA target genes in PMOV10 cells stimulated with ET-1 and transfected with si-p53, or si-TEAD for 72 h (**m**) or PMOV10 cells stimulated with ET-1 and treated with MAC or transfected with si-p53, or si-TEAD, or si-YAP, or si-β-arr1 for 72 h (**n**). Bars are means ± SD (*$p < 0.01$ vs CTR; **$p < 0.01$ vs ET-1; $n = 3$). **o**, **p** TEAD transcriptional activity (**o**) and ET-1 promoter activity (**p**) performed in PMOV10 cells stimulated with ET-1 and treated with MAC, and co-transfected with SCR, si-β-arr1, si-β-arr1 and mutant β-arr1Q394L-FLAG, rescued with β-arr1-FLAG, or in cells transfected with si-YAP, or si-TAZ, or si-p53, or si TEAD and TEAD-luc construct (**o**) or ET-1 promoter-luc construct (**p**) for 24 h. Bars are means ± SD (*$p < 0.01$ vs CTR; **$p < 0.001$ vs SCR ET-1; ***$p < 0.002$ vs β-arr1 silenced cells; ****$p < 0.05$ vs β-arr1 silenced cells treated with ET-1; $n = 3$). **q** Schematic representation of the transcriptional complex β-arr1/YAP/mutp53/TEAD bound to the specific ACATTCCA-box sequences on the promoters

levels, was more pronounced in combination treatment and even more in cells depleted of YAP and co-treated compared to those treated with the single treatment (Fig. 5e), demonstrating that ET-1R blockade by macitentan, impaired YAP-dependent cell survival and led to an enhanced sensitivity of HG-SOC cells to the cytotoxic agent. Interestingly, the strong setback in cell viability and enhanced levels of cleaved PARP (Fig. 5d, e), caused by macitentan and/or CIS treatments associated to β-arr1 or YAP depletion, was completely recovered by ectopic expression of β-arr1 or by the constitutively active YAP 5SA. This provides further evidence that cells over-expressing β-arr1 or YAP were more resistant to cisplatinum or combination treatment compared with parental cells. Overall these findings reveal the role of YAP as an effector of ET-1R/β-arr1 pathway in inducing apoptosis protection and that macitentan treatment, shutting down $ET_AR/β$-arr1/YAP-activity, reawakes cisplatinum sensitivity.

**Macitentan inhibits tumor metastasis and enhances platinum efficacy.** To evaluate the in vivo therapeutic potential of ET-1R

blockade in HG-SOC, we tested the ability of macitentan in controlling tumor growth in HG-SOC patient-derived xenografts (PDX). Of relevance, the hematoxylin-eosin analysis showed that the PMOV10 PDX recapitulated the histologic features of the HG-SOC patient (Supplementary Fig. 6a). In this HG-SOC pre-clinical model, we evaluated the following treatments: control (vehicle) versus macitentan (30 mg/kg/oral daily) and/or cisplatinum (8 mg/kg/i.p. once a week) in mono-therapy or in combination therapy. At the end of the treatment, we observed that macitentan induced tumor growth inhibition (Fig. 6a, left panels). Most importantly, a synergistic growth-inhibitory effect, as calculated by Chou and Talalay method, was observed when macitentan was used in combination with cisplatinum, compared with macitentan- or cisplatinum-treated mice (93% vs. 69% or 82% respectively). These findings provide strong in vivo evidence of the ability of macitentan to sensitize HG-SOC PDX to cisplatinum. Intriguingly, ChIP assays on nuclear extracts from tumors of PMOV10 PDX, showed the recruitment of β-arr1, YAP, mutp53 and TEAD on target gene promoters, such as *ANKRD1*

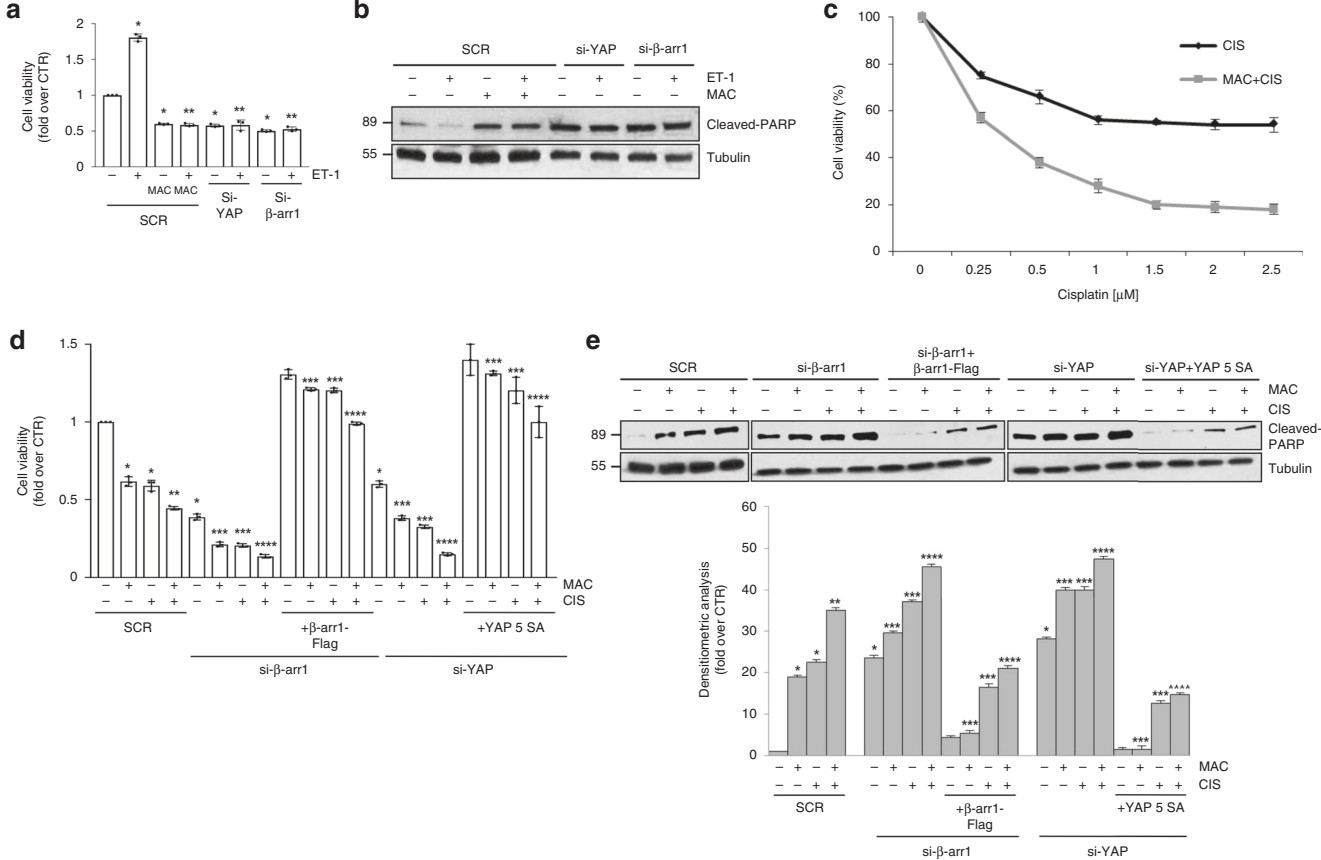

**Fig. 5** Macitentan impairs ET-1R/β-arr1/YAP-mediated cell survival and sensitize to platinum. **a** Effect on cell growth of PMOV10 cells stimulated with ET-1 and treated with MAC for 48 h or transfected with SCR, or si-β-arr1, or si-YAP. Bars are means ± SD (*$p < 0.01$ vs CTR; **$p < 0.01$ vs SCR ET-1) ($n = 3$). **b** IB analysis for cleaved-PARP protein expression in PMOV10 cells treated as in **a**. **c** Effect of different concentrations of cisplatinum combined or not with MAC (1 μM) after 48 h on cell vitality of PMOV10 cells. Data points are means ± SD ($n = 3$). **d** Effect of treatment with MAC and/or CIS and MAC + CIS for 48 h on cell growth of PMOV10 cells transfected with SCR, or si-β-arr1 or si-YAP and then rescued with β-arr1-FLAG or YAP 5SA-Myc, respectively. Bars are means ± SD; *$p < 0.001$ vs SCR CTR; **$p < 0.01$ vs SCR MAC or CIS; ***$p < 0.002$ vs CTR β-arr1 or YAP silenced cells; ****$p < 0.001$ vs β-arr1 or YAP silenced cells treated with MAC or CIS ($n = 3$). **e** IB analysis for cleaved-PARP protein expression in PMOV10 cells treated as in **d**. Bottom graph: quantification of cleaved-PARP normalized to tubulin. Bars are means ± SD; *$p < 0.0001$ vs SCR CTR; **$p < 0.001$ vs SCR MAC or CIS; ***$p < 0.0002$ vs CTR β-arr1 or YAP silenced cells; ****$p < 0.01$ vs β-arr1 or YAP silenced cells treated with MAC or CIS ($n = 3$)

and *ET-1* (Fig. 6b and Supplementary Fig. 6b–e). Notably, in vivo treatment with macitentan impaired this recruitment (Fig. 6b and Supplementary Fig. 6b–e). In line with the above data, in PMOV10 xenografted tumors macitentan and/or cisplatinum therapy induced a significant increase of pYAP protein expression levels, which was further enhanced upon combination therapy (Fig. 6c and Supplementary Fig. 6f). This indicated that macitentan might control tumor growth and enhances drug sensitivity through the cytoplasmic re-localization of YAP and the reduced recruitment of YAP on target gene promoters in vivo. To test the therapeutic anti-metastatic potential of macitentan, we evaluated its ability to control the intraperitoneal dissemination of PMOV10 (Fig. 6d, right panels) and OVCAR-3 (Fig. 6f, right panels) cells orthotopically implanted in nude mice. Macitentan treatment induced a significant inhibition in the number of tumor nodules, as well as treatment with cisplatinum. The effect was more significant in those mice treated in combination therapy (Fig. 6d, f) and was paralleled by significantly increased pYAP protein expression levels, as evidenced by immunoblotting analysis of tumor nodules of PMOV10 and OVCAR-3 xenografts (Fig.6e, g and Supplementary Fig. 6g). Collectively, the above data suggest that ET-1R blockade by macitentan impairs YAP/mutp53 transcriptional machinery; thereby representing an emerging anti-cancer therapeutic approach for combinatorial strategies for

HG-SOC harboring common p53 mutations to overcome compensatory mechanism of platinum-based therapy escape.

**ET-1/ET$_A$R and YAP expression as a prognostic signature in HG-SOC.** From TGCA analysis[32], it has been reported that different types of cancers have distinctive patterns of Hippo–p53 axis activation. Interestingly in tumors in which there is a strong pressure to mutate TP53, the expression of *CCNA*, along with *CTGF*, is significantly higher; thereby suggesting the existence of a transcriptionally active mutp53-YAP complex[56]. Based on this, we analyzed the mRNA levels of the two canonical YAP/mutp53 transcription target genes (*CTGF, CCNA*) in the same HG-SOC tissues ($n = 30$) displaying *YAP1* and *EDNRA* mRNA levels (Supplementary Fig. 7a-d). We evaluated the correlation in this cohort of HG-SOC patients of *CCNA, CTGF*, and *EDNRA* mRNA and we found a robust and statistically significant association between *CCNA, CTGF* and *EDNRA* (Fig.7a, b).

To strengthen the clinical relevance of the ET-1R signaling and YAP in HG-SOC patients, we analyzed the relationship between YAP, ET$_A$R and ET-1 protein expression in HG-SOC tissues ($n = 21$) and in normal ovarian tissues ($n = 6$). We found that the expression YAP, ET$_A$R and ET-1 in normal ovarian tissues derived from ovariectomy were lower and more homogeneous

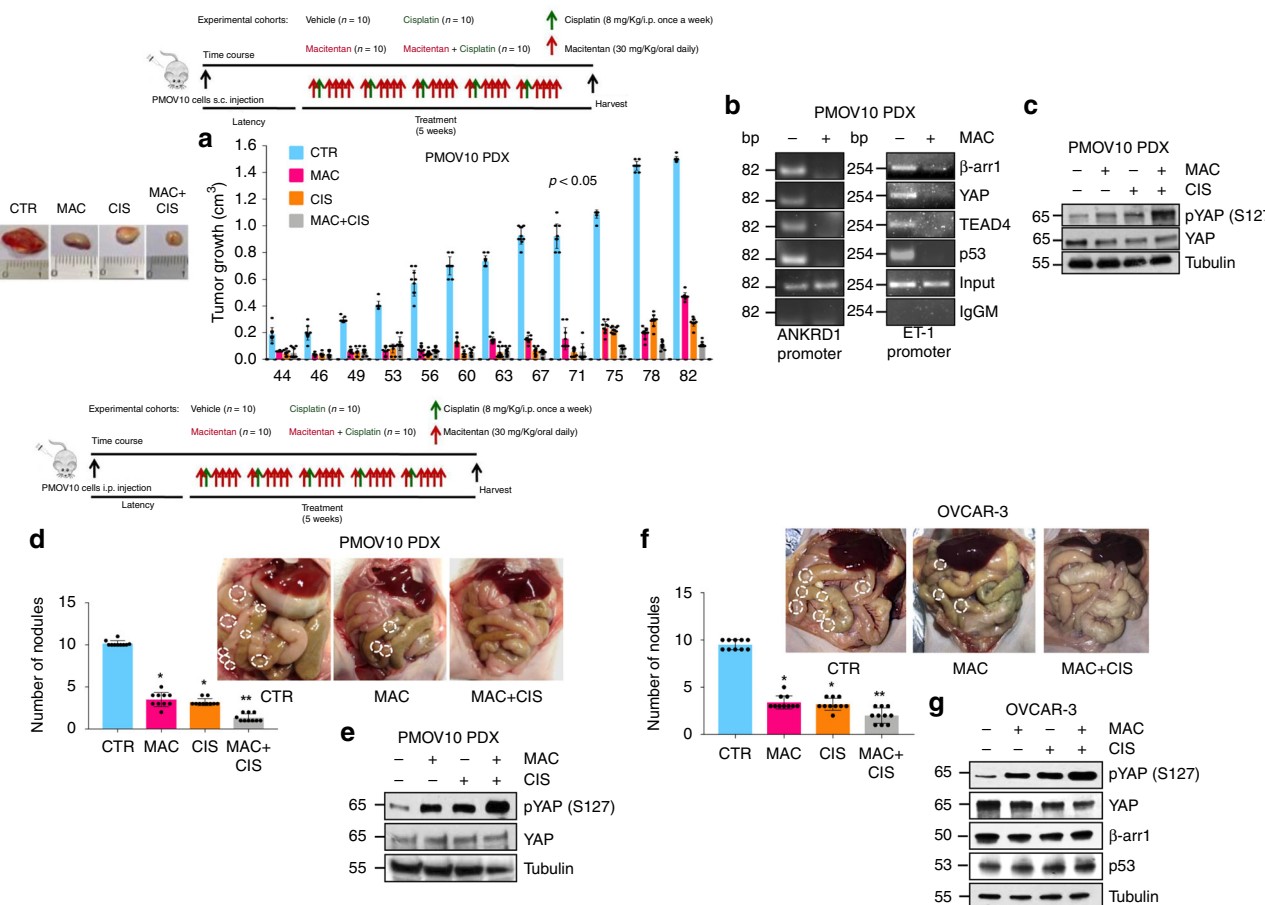

**Fig. 6** Macitentan inhibits metastases sensitizing to platinum. **a** Patient-derived xenografts (PDX), obtained transplanted subcutaneously PMOV10 cells in nude mice, were treated with vehicle (CTR) or MAC (30 mg/Kg/day, oral daily), CIS (8 mg/kg/i.p. once a week) in mono-therapy or with MAC + CIS combination therapy (upper, treatment scheduling). The comparison of tumor growth levels by two-way ANOVA with group-by-time interaction was statistically significant ($p < 0.05$). Bars are means ± SD of ten mice for group. *Left*. Representative tumors of PMOV10 PDX. **b** The binding of β-arr1, YAP, TEAD, and p53 to *ANKRD1* and *ET-1* promoter regions was analysed in PDX tumors treated or not with MAC by ChIP assays followed by PCR. **c** pYAP (S127) and YAP protein expression was evaluated by IB analysis in PDX tumors of PMOV10 treated with MAC and/or CIS and MAC + CIS. Tubulin was used as loading control. **d, f** Nude mice intraperitoneally (ip) injected with PMOV10 (**d**) or OVCAR-3 (**f**) cells and treated as in a (upper, treatment scheduling). Bars are means ± SD of ten mice for group. (\*$p < 0.001$ vs CTR; \*\*$p < 0.01$ vs MAC or CIS). At the end of the treatment ip organs were examined for tumor nodules. Right, representative ip nodules of PMOV10 PDX and OVCAR-3 xenografts were indicated by white dotted-line circles. **e, g** pYAP (S127) and YAP protein expression of PMOV10 PDX ip nodules extracts (**e**) or pYAP, YAP, β-arr1, and p53 protein expression was evaluated by IB analysis of ip nodules extracts of OVCAR-3 xenografts (**g**) treated with MAC and/or CIS and MAC + CIS. Tubulin was used as loading control

compared to those from HG-SOC tissues (Fig. 7c). These findings suggest that YAP overexpression is associated with ET_AR/ET-1 axis in a high percentage of HG-SOC compared to normal ovarian tissues, reflecting the close network between the ET_AR and YAP pathways in HG-SOC. In line with these results, we analyzed the YAP, ET_AR and ET-1 protein expression in clinical specimens from the human protein atlas. We found that YAP, ET_AR, and ET-1 displayed strong expression in HG-SOC, compared to ovarian tissues (Supplementary Fig. 7e). In addition, we evaluated the combined expression of ET_AR, β-arr1 and YAP as prognostic gene signature. By using TCGA data-set, we found that high *ARRB1-EDNRA-CYR61-CTGF-ANKRD1* mRNA expression significantly correlated with a worse prognosis, expressed as OS [HR = 1.54 (1.1–2.15), $P = 0.012$] and DFS [HR = 1.58 (1.13–2.22), $P = 0.007$] (Fig.7d). Similarly, TCGA data set disclosed that high *YAP1-ARRB1-EDNRA* mRNA expression significantly correlated with a worse prognosis, expressed as overall survival (OS) [hazard ratio (HR) = 1.37 (1.1–1.7), $P = 0.006$] and disease free survival (DFS) [HR = 1.34 (1.1–1.67), $P = 0.009$] (Fig. 7e). Overall these results, showing

that the expression of the network-based YAP target gene/*ARRB1/EDNRA*, or *YAP1/ARRB1/EDNRA*, is associated with poor clinical outcome compared to those with lower expression, suggest that YAP signature in association with *EDNRA* and *ARRB1* expression might be especially valuable for the prognosis of recurrent mutp53 HG-SOC patients.

## Discussion

The integration of diverse signaling pathways operating concomitantly within tumor cells could represent in HG-SOC a strategy to escape chemotherapy response. Therefore the identification of interconnected pathways might allow the discovery of potential druggable targets to develop more effective therapies. In this study we identify the oncogenic Hippo transducer YAP, as a major regulator to elicit ET-1/ET-1R signaling into a specific transcriptional program, pivotal for enabling HG-SOC survival and invasive behavior. To integrate these two signaling pathways we identified the multifunctional scaffold protein β-arr1 that controls spatiotemporally the activity of interacting proteins

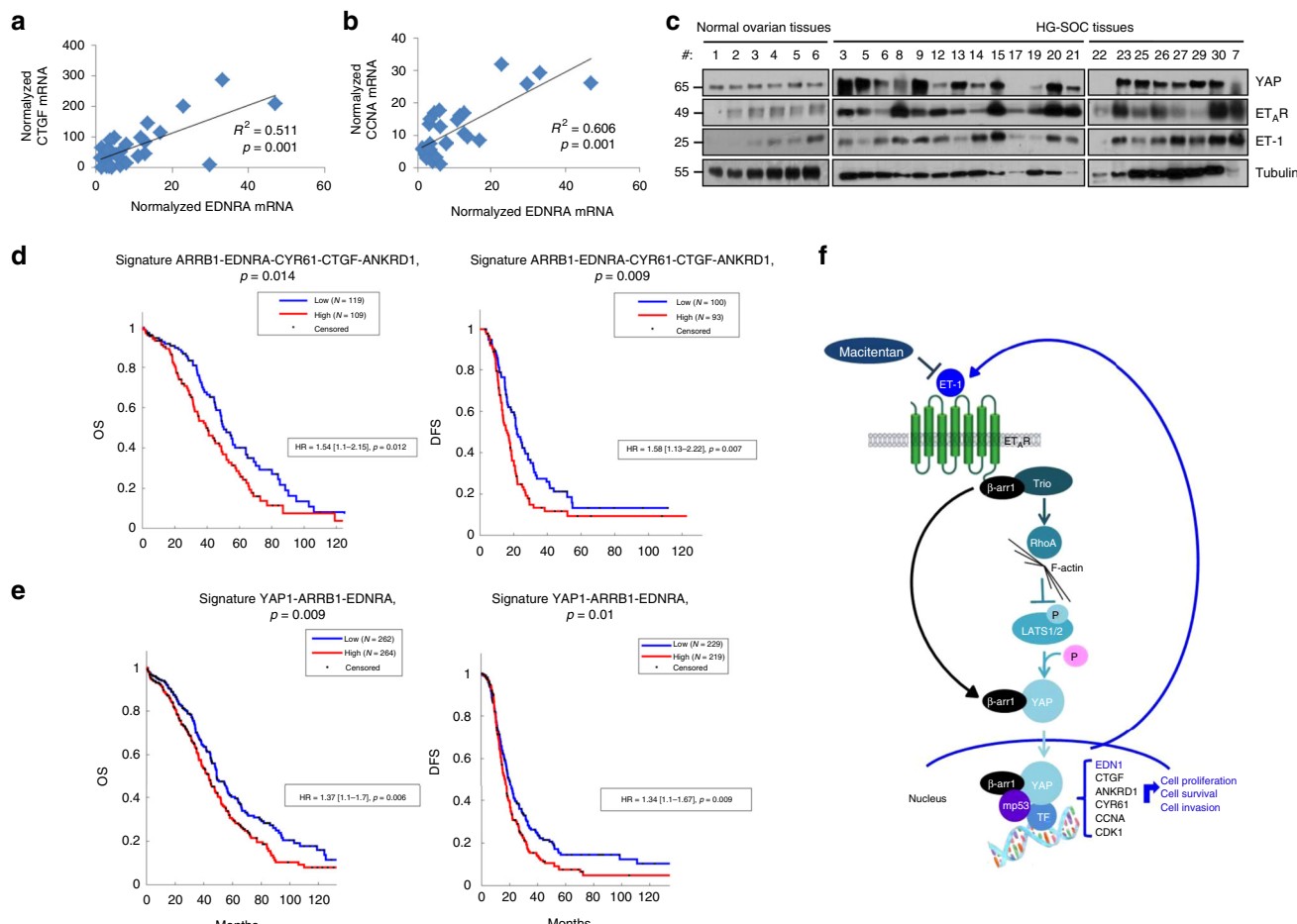

**Fig. 7** Expression of YAP and ET-1/ET$_A$R as a prognostic signature in HG-SOC patients. **a**, **b** Relative *EDNRA* (ET$_A$R), *CCNA*, and *CTGF* mRNA expression levels in 30 HG-SOC human specimens normalized for *CYPA* mRNA expression were analyzed for their correlation: **a** *EDNRA* and *CTGF* correlation, **b** *EDNRA* and *CCNA* correlation. **c** YAP, ET$_A$R, and ET-1 expression was evaluated by IB analysis of 21 HG-SOC human specimens and six normal ovarian tissues. Tubulin was used as loading control. **d**, **e** Overall survival (OS) and disease free survival (DFS) of HG-SOC patients with high (z score > 0.5) and low (z score < 0.5) combined expression levels of ET$_A$R, β-arr1 correlated with YAP gene signature (*CTGF, ANKRD1* and *CYR61*) (p < 0.014 for OS; p < 0.009 for DFS) (**d**). **e** OS and DFS of HG-SOC patients with high (z score > 0) and low (z score < 0) combined signature of ET$_A$R, β-arr1, and YAP (p < 0.009 for OS; p < 0.01 for DFS). Survival analyses from TCGA data set were evaluated by Kaplan–Meier method and a log rank test was used to establish the statistical significance of the distance between curves. High and low gene expression values were defined basing on the z-scores of the signals. **f** Schematic model of ET$_A$R-dependent YAP activation in HG-SOC models. Mechanistically, nuclear β-arr1 binds mutp53/YAP/transcriptional factor complex to regulate target gene expression. In parallel, ET-1 activates RhoA GTPase, actin dynamics and LATS, for YAP nuclear localization in HG-SOC, through a non-canonical β-arr1/Trio pathway. Integrating these important routes, β-arr1 appears as an emerging hub node, which characterizes important properties that control aggressive features of HG-SOC. Blunting ET-1R/β-arr1-mediated signaling by using macitentan may impair mutp53/YAP/transcriptional machinery, inhibiting tumor growth, metastasis, and rewiring HG-SOC to survive to chemotherapy attack

involved in a plethora of cellular functions. Using different approaches, we provide evidence for the required role of β-arr1 in ET$_A$R-regulated assembly of β-arr1/YAP nuclear complex that mediates transcriptional program leading to tumor growth, metastasis and reduced platinum susceptibility in HG-SOC (Fig. 7f). In parallel, we establish ET-1 as an external cue that activates a route, including RhoA GTPase activity, for YAP nuclear localization in HG-SOC, through an unforeseen β-arr1/Trio physical interaction (Fig. 7f).

The interaction with diverse sets of partner positions β-arr1 as critical regulator of GPCR signal transduction, which allows to integrate GPCR-mediated signals with other inputs. Besides β-arr1 we found that YAP enrolls mutp53 in a trimeric nuclear complex, providing evidence that nuclear β-arr1 is part of a functional YAP/mutp53-mediated transcriptional complex that aberrantly tunes the expression of YAP target genes, such as

*CYR61*, *CTGF*, *ANKRD1*, *CCNA*, and *CDK1*. Notably, it also controls *EDN1*, thereby amplifying a positive feed-back loop that delivers ET-1 to the ET$_A$R expressed in HG-SOC cells. Such a positive feed-back loop would be important to sustain a persistent and unscheduled YAP activity, enhancing cell survival and invasion in HG-SOC. Moreover, our findings point toward a role for β-arr1/YAP/mutp53 as a major nexus that integrates and decodes different stimuli into precise transcriptional programs in HG-SOC upon ET$_A$R activation. The data presented here describe a mechanism by which ET-1R/β-arr1, engages the oncogenic cross-talk between mutp53 and YAP, amplifying aberrantly YAP activity in HG-SOC. It might also be that β-arr1 can anchor different transcription factors, as we report for NFY, to expand its own agenda providing a selective advantage to HG-SOC cells. Hence, our study provides mechanistic insights by which β-arr1-mutp53-YAP complex represents the initial scaffold on

which transcriptional regulatory networks could be built to regulate different biological outcomes. It has been increasingly clear[34,57–60] that the interactions with active transcription factors, tethering mutp53 to the DNA, can mediate and broaden the ability of gain of function mutp53 proteins to regulate gene expression. Further studies should attempt to clarify whether ET-1, through distinct β-arr1-complexes with YAP and mutp53 can recruit or interact with factors, such as HIF-1α, β-catenin, or NF-κB, to orchestrate interconnected pathway network regulating tumor growth and progression in HG-SOC, and in other types of cancer harbouring gain of function TP53 mutations.

Consistent with recent studies that identified RhoA-induced signaling as important mediator of OC short survival, invasion and apoptosis resistance[61–63] and that RhoA may stabilize mutp53 for its oncogenic functions[64], our findings, demonstrating the ability of ET$_A$R/β-arr1 to activate Trio/RhoA-induced YAP/mutp53 nuclear accumulation, provide evidence for a further layer for mutp53/YAP regulation in HG-SOC. In such context, mutp53 protein is a downstream effector of ET-1R/β-arr1 axis. The exploration of other inputs, as ET-1R/β-arr1, may provide insights into actionable nodes for HG-SOC treatment. In this regard, our study proposes a selective β-arr1-biased ligand antagonist that interfering with ET-1R/β-arr1-driven YAP activation impairs OC growth, progression and apoptosis evasion to chemotherapy. Thus, the FDA-approved dual ET-1 receptor antagonist, macitentan, preventing the β-arr1-dependent network formation, interferes with YAP cytoplasmic-nuclear shuttling, disrupts YAP/mutp53 transcriptional activity, and sensitizes HG-SOC cells to platinum-induced apoptosis. Given that TP53 is frequently mutated in HG-SOC, combining an agent able to interfere with mutp53-mediated oncogenic activity, such as macitentan, with cisplatinum therapy may represent a valuable therapeutic option that warrants clinical exploration. The efficacy of this co-therapy, through the blockade of ET-1R/β-arr1 network, can re-enable apoptosis in HG-SOC cells, rendering them more vulnerable to cisplatinum. The therapeutic benefit of macitentan is to target not only HG-SOC cells expressing ET$_A$R, but also to interfere with tumor microenvironmental elements (TME), such as fibroblasts, vascular, lymphatic, and immune cells, which mainly expressed ET$_B$R[3,4,36,65,66], representing a therapeutic strategy which may be used to design a targeted therapies towards TME elements and HG-SOC cells in combination with chemotherapeutics for cancer patients still suffering from limited treatment options.

Of clinical relevance, in our study, we found that, along with ET-1/ET$_A$R axis, YAP expression was up-regulated in HG-SOC tissues compared to normal ovarian tissues. Additionally, HG-SOC patients harboring TP53 mutations, with a combined high expression levels of ET-1R/β-arr1/YAP have poor prognosis compared to patients who lack this network-based signature, further highlighting the worse outcomes generated by the integration between ET-1R/β-arr1 and YAP pathways. These findings contribute to identify a predictive gene expression signature for recurrent HG-SOC patients. Overall, the present study supports the attractive opportunity to interfere with the interconnection between ET-1R/β-arr1 and YAP pathways, which is intertwined with mutp53, disclosing the clinical implication of this druggable pathway.

## Methods

**Cell cultures and reagents**. HG-SOC primary cells were obtained from freshly-isolated ascitic fluid of HG-SOC patients undergoing surgery for ovarian tumor by laparotomy or paracentesis at the Gynecological Oncology of our Institute. The study protocol for tissue collection and clinical information was approved by the institutional review board (IRB) and patients provided written informed consent authorizing the collection and use of the tissue for study purposes. Briefly, cells were harvested by centrifugation at $200 \times g$ for 5 min at room temperature,

resuspended in Dulbecco's PBS, and then centrifuged through Ficoll-Histopaque 1077 (Sigma-Aldrich, St. Louis, Missouri,USA). Interface cells were washed in culture medium, and $5 \times 10^6$ viable cells were seeded in 75-cm$^2$ culture flasks, in RPMI 1640 (Gibco, Grovemont Cir, Gaithersburg, USA) containing 1% penicillin-streptomycin and 10% fetal bovine serum. All experiments were conducted between the first and second in vitro passage. The purity of primary cultures was assessed by immunophenotyping with a panel of monoclonal Abs (including WT1, keratin 7, calretinin and OCT-125) recognizing ovarian tumor-associated antigens by the alkaline phosphatase-peroxidase-antiperoxidase method. The early passage HG-SOC patient-derived cell line is named PMOV10 where PM stands for Pre-clinical Models, OV stands for ovarian serous cancer, and # is the order in which the cell line is established. PMOV10 (TP53 mutant R337T, Representative Visualization in IGV, Supplementary Fig. 1b), closely recapitulates the histologic and molecular features of HG-SOC patient (stage III, age 69, Supplementary Fig. 6a). PMOV10 and MDA-MB-468 cells were cultured in RPMI 1640 (Gibco) containing 1% penicillin-streptomycin and 10% fetal bovine serum, OVCAR-3 cells were cultured in RPMI-1640 containing 1% penicillin-streptomycin and 10% fetal bovine serum and 1X non-essential amino acid (MEM) (Gibco), under a humidified atmosphere of 5% CO2 at 37 °C. OVCAR-3 (HTB-161) and MDA-MB-468 (HTB-132) were obtained from American Type Culture Collection (ATCC). Cell lines were tested for mycoplasma and validated by short tandem repeat (STR) profiling. Before each experiment, cells were serum starved by incubation in serum-free medium for 24 h. ET-1 was used at 100 nM and was purchased from Bachem (Bachem, Bubendorf, Torrance, Switzerland). Macitentan, also called ACT-064992 or N-(5-[4-bromophenyl]-6-{2-[5-bromopyrimidin-2-yloxy]ethoxy}pyrimidin-4-yl)-N'-propylsulfamide, was added 30 min before ET-1 at a dose of 1 μM and was kindly provided by Actelion Pharmaceuticals, Ltd. (Actelion Pharmaceuticals, Allschwil, Switzerland) BQ123 and BQ788 were used at 1 μM and were purchased from Bachem. Lat B and CYTO D were both purchased from Sigma-Aldrich and respectively used at 2 μM for 30 min and 1 μM for 4 h. Y-27632 was used at 10 μM for 1 h and was purchased from Alexis Corporation (Alexis Corporation Lausen, Switzerland). CT04 was used at 2 μg/ml for 4 h and purchased from Cytoskeleton. Inc (Cytoskeleton. Inc, Denver, USA). Cisplatin (CIS) was used at 1 mg/ml and was purchased from Teva (TEVA, Petach Tikva, Israel).

**Sequencing and variant call**. TP53 mutation analysis was performed on genomic DNA (gDNA) extracted from PMOV10 cells using the extraction buffer (10 mM TrisCl pH 8.0, 0.1 M EDTA pH 8.0, 20 μg/ml RNAse, 0.5% SDS). After quantification with Qubit dsDNA HS Assay Kit (Termo Fisher Scientific, Waltham, USA), Fifty nanograms of gDNA were used for library preparation with a custom HaloPlex HS panel (Agilent Technologies, Santa Clara, California, USA) targeting TP53 coding exons of the canonical p53 isoform (RefSeqNM000546, https://www.ncbi.nlm.nih.gov/refseq/) and non-canonical isoforms comprising that one including the exon 9 (ENST00000359597), which contains R337T mutation. The library was finally sequenced on MiSeq platform (Illumina, San Diego, California, USA), in paired-end mode (2 × 150 bp), using MiSeq Reagent Kit v2 (Illumina). Raw sequencing data were preprocessed to remove low quality, amplicon footprints and technical artifacts with BBDUK (BBMap - Bushnell B. - sourceforge.net/projects/bbmap/). Alignment of sequencing reads to human genome reference hg19 (GRCh37 NCBI) was performed with BWA 0.7.12. Duplicated reads were marked with Agilent LocatIt 3.5.1.46 (Agilent Genomics NextGen Toolkit (AGeNT), https://www.genomics.agilent.com/it/NGS-Data-Analysis-Software/AGeNT/). Small nucleotide variants and small indels were called with Freebayes 1.1 with default parameters. Variants were filtered and annotated with SNPSift (https://doi.org/10.4161/fly.19695) on dbSNP (https://www.ncbi.nlm.nih.gov/projects/SNP/), Exac (http://exac.broadinstitute.org/), COSMIC (https://cancer.sanger.ac.uk/cosmic) and GnomAD (http://gnomad.broadinstitute.org/). Variants effects were predicted using SNPEff (https://doi.org/10.4161/fly.19695). Newly discovered variants were visually inspected with IGV 2.3.92 (https://software.broadinstitute.org/software/igv/). The observed genomic nucleotide variant is a C to G, on chromosome 17, position 7569546 (hg19). Non silent amino acid change, as reported by SNPEff for the Ensembl (https://www.ensembl.org/) gene ENSG00000141510, Ensembl transcript ENST00000359597, TP53-019: coding sequence variant c.1010 G > C, amino acid change p.Arg337Thr, effect prediction "Missense Variant". Variant depth 290x (769x before duplicate removal), estimated Variant Allele Frequency (VAF) 56%.

**DNA copy number analysis**. Total RNA was isolated using the Trizol (Life Technologies, Carlsbad, California, USA) according to the manufacturer's protocol. RNA was reversed transcribed using the SuperScript® VILO™ cDNA synthesis kit (Life Technologies). The ET-1, ET$_A$R and β-arr1 copy number analysis was evaluated by quantitative real-time-PCR, performed by using Light Cycler rapid thermal cycler system (Roche Diagnostics, Basilea, Switzerland) and Light Cycler-Fast Start DNA Master Plus SYBR Green mix (Roche Diagnostics). The number of each gene-amplified product was normalized to the number of cyclophilin-A (CYPA) amplified product. The primers employed for DNA copy number analysis are listed below. The experiments were performed in triplicates.

**Ectopic expression and silencing**. For exogenous expression of β-arr1, we used pcDNA3-β-arr1-FLAG (wild-type) plasmid construct, a 'wobble' mutant construct encoding rat β-arr1 sequences resistant to small interfering RNA targeting kindly provided by Dr Robert Lefkowitz (Howard Hughes Medical Institute, Duke University, Durham, NC, USA). Mutation of Gln-394 in Leu of β-arr1-FLAG (β-arr1Q394L-FLAG) construct was done by using Quick Change II XL Site-Directed Mutagenesis Kit (Agilent Technologies). All constructs were verified by sequencing. For exogenous expression of a constitutive active form of YAP, we used pQCXIH-Myc-YAP-5SA plasmid construct (Addgene plasmid # 33093)[67]. For transient expression of full-length β-arr1 (AU5-β-arr1) or of the β-arr1 deletion mutant (AU5-β-arr1-1-180N)[14,46], we used constructs tagged with an AU5 epitope at their carboxyl termini kindly provided by Professor Richard D Ye (Department of Pharmacology, College of Medicine, University of Illinois, Chicago, IL, USA). For transient expression of β-arr1-FLAG, or β-arr1Q394L-FLAG, or pQCXIH-Myc-YAP 5SA, or AU5-β-arr1, or AU5-β-arr1-1-180N constructs we used LipofectA-MINE 2000 reagent (Life Technologies) following the manufacturer's instructions. Cells transfected with the empty vectors pCDNA3 or pQCXIH were used as control (MOCK).

Silencing of β-arr1, $ET_AR$, $ET_BR$, Trio, YAP, TAZ, TEAD4, and GNAQ (Gαq/11) were performed by using Lipofectamine RNAiMAX Reagent (Life Technologies) according to manufacturer's instructions by using the following ON-TARGET plus SMART pool siRNAs (Dharmacon, Lafayette, Colorado, USA): ARRB1: L-011971-00-0050, ETAR: L-005485-00-0050, ETBR: L-003657-00-0020, TRIO: M-005047-00-0020, YAP1: L-012200-00, WWTR1 (TAZ): L-016083-00, TEAD4: L-019570-00 and GNAQ (Gαq/11): L-008562-00. SiGENOME Control Pool Non-targeting was used as negative control (SCR). RhoA silencing was performed by using the Silencer Select siRNA for RhoA (Ambion). For human LATS1 we performed siRNA oligonucleotides transfection by using the following sequence[68]: 5′-CAUA CGAGUCAAUCAGUAA-3′. For human p53 silencing we performed siRNA oligonucleotides transfection by using the following sequence[69]: 5′-GACUCCAGUG GUAAUCUAC-3′. To confirm efficient knockdown, total cell lysates were collected at the end point of each experiment and analyzed by immunoblotting.

**Immunoblotting and immunoprecipitation**. NE-PER nuclear and cytoplasmic extraction reagents kit (Thermo Scientific, Waltham, Massachusetts, USA) was used to separate cytoplasmic and nuclear fractions. Whole cell lysates were prepared using a modified RIPA buffer (50 mM Tris-HCl pH 7.4, 250 mM NaCl, 1% Triton X-100, 1% sodium deoxycholate, 0.1% SDS) containing a mixture of protease and phosphatase inhibitors. Protein content of the extracts was determined using Bio-Rad Protein Assay Kit (Bio-Rad, Hercules, California, USA). IB using anti-PCNA (1:200, F-2, cat. no. sc-25280, Santa Cruz Biotechnology, Dallas, Texas, USA) and anti-tubulin (1:200, DM1A cat. no. sc-32293, Santa Cruz Biotechnology) Abs were used as loading control and to assess the purity of the nuclear and cytoplasmic fractions, respectively. Whole cell lysates or separated fractions were resolved by SDS/PAGE. The membranes were blocked in TTBS (TBS with 0.1% Tween 20) containing either 5% dry milk or BSA. Primary Abs incubations were performed in TTBS with either 5% dry milk or BSA overnight at 4 °C. After washing, the membranes were incubated with the appropriate secondary peroxidase conjugated Abs for 1 h in TTBS with either 5% dry milk or BSA. For IP, precleared whole cell lysates or cytoplasmic and nuclear cell fractions were incubated with indicated Abs or with anti-rabbit IgG Isotype Control (Life Technologies), anti-mouse IgG Isotype Control (Life Technologies), and anti-sheep IgG Isotype Control (Life Technologies) and protein G-agarose beads (Santa Cruz Biotechnology) at 4 °C overnight. For the IP assays, the nuclear extracts were treated with 15 u/ml of DNase I (Life Technologies). The IP and input (3% of the total nuclear extracts) samples were boiled for 5 min in SDS loading buffer, loaded onto pre-casted 7,5, 10 or 4–20% SDS/PAGE (Bio-Rad), and transferred by using Trans-Blot transfer pack (Bio-Rad) and IB with different Abs as before. To obtain clean and specific IB signals of β-arr1 and p53, which run very close to heavy chain of IgG, we used HRP-conjugated protein A peroxidase (1:5000, cat. no. 32400, Pierce) instead of HRP-conjugated secondary Abs. Blots were developed with the enhanced chemiluminescence detection system (Clarity Western ECL Substrate Bio-Rad) and quantified using NIH image program (Image J). All antibodies used in immunoblotting and immunoprecipitation assays are listed in Supplementary Table 1. Uncropped scans of immunoblots are provided in Supplementary Figs. 8–15.

**Rho GTPase activation assay**. RhoGTP levels were assessed using a Rho-binding domain (RBD) affinity precipitation assay (Cytoskeleton, Inc.). Briefly, cells were lysed in 300 μl of ice-cold MLB lysis buffer (25 mM 4- (2-hydroxyethyl)-1-piper-azineethanesulfonic acid, 150 mM NaCl, 1% Nonidet P-40, 10 mM MgCl2, 1 mM EDTA, 10% glycerol, and 0.3 mg/ml phenylmethylsulfonyl fluoride complemented with protease inhibitors and 1 nM sodium orthovanadate). Glutathione Stransferase (GST)–Rhotekin coupled to glutathione agarose was added to each tube, and samples were rotated at 4 °C for 60 min. Beads were washed, and proteins were eluted in 25 μl of 2x Laemmli (Bio-Rad) reducing sample buffer by heating to 95 °C for 5 min. Detection of Rho-GTP was performed by IB analysis using anti-RhoA-B-C (1:500, clone55, cat. no. 05–778, Millipore, Burlington, Massachusetts, USA) Ab or specific anti-RhoA (1:1000, Cytoskeleton, Inc.) Ab.

**Gelatin zimography**. Cell supernatants were electrophoresed for analysis in 9% SDS-PAGE gels containing 1 mg/ml gelatin. The gels were washed for 30 min at 22 °C in 2.5% Triton X-100 and then incubated in 50 mM Tris (pH 7.6), 1 mM ZnCl2, and 5 mM CaCl2 for 18 h at 37 °C. After incubation the gels were stained with 0.2% Coomassie Blue. Enzyme-digested regions were identified as white bands on a blue background. The experiments were performed in triplicates.

**Immunofluorescence staining**. Cells were fixed in 4% formaldehyde for 20 min at room temperature. Cells were then washed with PBS twice, permeabilized in 0.3% Triton X-100 in PBS for 5 min and blocked in PBS/0,5% BSA for 60 min at room temperature. After cells were incubated overnight at 4 °C with anti-YAP (1:150, G-6, cat. no. sc-376830, Santa Cruz Biotechnology) or with anti-p53 (1:50, DO-1, cat. no. sc-126, Santa Cruz Biotechnology) both diluted in PBS/1% BSA. Next day, Alexa Fluor 488-labeled goat anti-mouse (1:250, cat. no. A11001, Life Technologies) and goat-anti-rabbit (1:250, cat. no. A11008, Life Technologies) for YAP and p53 respectively, were added as secondary antibodies for 2 h at room temperature. DAPI (Bio-Rad) was used for nuclear counterstain for 15 min at room temperature. Images of representative cells for each labeling condition were captured at ×40, ×64, and ×100 magnitudes with a Leica DMIRE2 deconvolution microscope equipped with a Leica DFC 350FX camera and elaborated by FW4000 deconvolution software (Leica, Wetzlar, Germany). The experiments were performed in triplicates.

**RNA extraction and quantitative real-time PCR (qRT-PCR)**. Total RNA was isolated using the Trizol (Life Technologies) according to the manufacturer's protocol. RNA was reversed transcribed using the SuperScript® VILO™ cDNA synthesis kit (Life Technologies). The expression of CYR61, CTGF, ANKRD1, EDN1, CCNA, CDK1, and CYPA mRNA was evaluated in the 7500 Fast Real-Time PCR System Mix (Applied Biosystems, Foster City, California, USA), using Power SYBR Green PCR Master Mix (Applied Biosystems). The levels of gene expression were determined by normalizing to CYPA mRNA expression and expressed in relative mRNA level ($2^{\wedge}\Delta\Delta ct$). The values obtained from different experiments were averaged, and data are presented as means ± SD. The primers employed for real-time PCR are listed in Supplementary Table 2.

**Chromatin immunoprecipitation**. Chromatin was extracted from $5 \times 10^6$ cells. Briefly, cells were crosslinked with 1% formaldehyde for 8 min at room temperature. Chromatin was sheared by sonication, centrifuged and diluted in 50 mM Tris pH 8.0, 0.5% NP-40, 0.2 M NaCl, 0.5 mM EDTA. One-twentieth of the precleared chromatin was used as the input for the Chromatin immunoprecipitation (ChIP) assay. The precleared chromatin was rotated overnight with primary Ab or IgG. The primary antibodies used were as follows: anti-β-arr1 (2 μg/μl, E274, cat. no. ab32099, Abcam), anti-YAP (2 μg/μl, H-125, cat. no. sc-15407, Santa Cruz Biotechnology), anti-TEAD4 (2 μg/μl TEF-3, N-G2, cat. no. sc-101184, Santa Cruz Biotechnology), anti-p53 (2 μg/μl, DO-1, cat. no. sc-126, Santa Cruz Biotechnology), anti-NFY, anti-rabbit IgG Isotype Control (Invitrogen), anti-mouse IgG Isotype Control (Invitrogen) and anti-sheep IgG Isotype Control (Invitrogen). Immune complexes were extracted and the purified DNA was examined by PCR or qReal-Time PCR. The primers used are listed in Supplementary Table 2. The experiments were performed in triplicates.

**Luciferase reporter gene assay**. To measure the transcriptional activity of TEAD, cells transiently silenced with SMART pool siRNA for β-arr1, YAP, TAZ, TEAD and p53, and/or with non-targeting siRNA as negative control (Dharmacon), using Lipofectamine RNAiMAX Reagent (Life Technologies), were transiently co-transfected with 1 μg of synthetic TEAD luciferase reporter plasmid, containing four TEAD binding sites, 100 ng pCMV-β-galactosidase (Promega) vector, pcDNA3–β-arr1–FLAG vector and β-arr1–Q394L–FLAG vector by using Lipo-fectAMINE 2000 reagent (Invitrogen). After 24 h of transfection, serum-starved cells were treated with ET-1 and/or macitentan for additional 24 h. Reporter activity was measured using the Luciferase assay system (Promega, Madison, Wisconsin, USA) and normalized to β-galactosidase activity. To measure the ET-1 promoter activity, cells transiently silenced with SMART pool siRNA for β-arr1, YAP, TAZ, TEAD and p53, and/or with non-targeting siRNA as negative control (Dharmacon), using Lipofectamine RNAiMAX Reagent (Life Technologies), were transiently co-transfected with 1 μg of ET-1 promoter luciferase reporter plasmid, and 100 ng pCMV-β-galactosidase (Promega) vector, pcDNA3–β-arr1–FLAG vector and β-arr1–Q394L–FLAG vector, using LipofectAMINE 2000 reagent (Life Technologies). After 24 h of transfection, serum-starved cells were treated with ET-1 and/or macitentan for additional 24 h. Reporter activity was measured using the Luciferase assay system (Promega) and normalized to β-galactosidase activity. The experiments were performed in triplicates for all conditions described.

**Cell viability analysis**. HG-SOC PMOV10 cells were seeded in triplicates, in 24-well plates. The cells were transiently transfected with si-YAP or si-β-arr1 or with a non-targeting siRNA or with the β-arr1-FLAG vector or YAP 5SA-Myc vector or with the empty vector (MOCK) and treated with ET-1, macitentan and cisplatinum, alone or in combination. After 48 h cell viability was determined by counting cells, for each time point, using a Neubauer-counting chamber and a bright field

miscroscope. The trypan blue dye exclusion method was used to evaluate the percentage of viable cells. The experiments were performed in triplicates for all conditions described.

**Chemoinvasion assay**. Chemoinvasion assays were carried out using BioCoat growth factor reduced Matrigel Invasion Chamber (BD Biosciences, Franklin Lakes, New Jersey, USA). Cells ($0.5 \times 10^5$) transiently silenced for YAP, TAZ, β-arr1, and p53, were stimulated with serum-free medium alone or with ET-1 and/or macitentan, added to the lower chamber. The cells were left to invade for 12 h at 37 °C. Cells on the upper part of the membrane were scraped using a cotton swab and the invading cells were stained using Diff-Quick kit (Dade Behring, Deerfield, Illinois, USA). The experiments were performed in sextuplicates. From every transwell, several images were taken under a phase-contrast microscope at ×4 magnification and two broad fields were considered for quantification. Cells were counted by using Image J program. The results of the analysis of the individual photos are depicted as dots in the various graphs, normalized to control and shown as fold of control and quantified using NIH image program (Image J).

**Immunohistochemistry**. IHC of human HG-SOC sections, from which HG-SOC PMOV10 cells derived, was performed by using anti-p53 (1:800, DO-07, cat. no. NCL-L-p53-DO7, Leica) and anti-YAP (1:400, 1A12, cat no. 12395, Cell Signaling, Danvers, Massachusetts, USA) Abs. The avidin-biotin indirect immunoperoxidase staining was performed using the Vectastain Elite kit (Vector Laboratories, Burlingame, California, USA). Sections in which the incubation with the primary Ab was substituted by isotype-matched IgG were used as control. AEC was used as chromogenic substrate and Mayer's haematoxylin as nuclear counterstain.

**Patient-derived xenograft drug response assays**. Athymic (nu+/nu+) female mice, 5- to 6-week of age (Charles River Laboratories, Milan, Italy), were subcutaneously injected with $1.8 \times 10^6$ PMOV10 cells, in 200 μl PBS. All the animal experiments were performed in accordance with the Italian Ministry of Health guidelines after approval by the Animal Welfare Body of Regina Elena Cancer Institute of Rome and comply with all relevant ethical regulations. When the tumors became detectable (44 days after tumor injection, engraftment rate 70%), we randomly divided the mice into four groups ($n = 10$), undergoing the following treatments: control (CTR; vehicle) versus macitentan (MAC; 30 mg/kg/oral daily) and/or CIS (8 mg/kg/i.p. once a week) in mono-therapy or in combination therapy. Tumor volume was measured with caliper and the tumor growth curve was plotted. Tumor volume was calculated using the formula: π/6 larger diameter × (smaller diameter)2. At the end of the treatment (5 weeks), all mice were euthanized and tumor xenografts were extracted. A section of the tumor xenograft was formalin-fixed, and paraffin-embedded. Haematoxylin-eosin-stained sections were subsequently evaluated by a pathologist in comparison with patient tumor. The tumor xenografts were harvested and preserved for further analysis. Values represent the mean ± SD of ten mice in each group from two independent experiments. For metastasis assays $2.5 \times 10^6$ viable PMOV10 or OVCAR-3 cells were intraperitoneally injected into female athymic nude mice. Two weeks after cell injection, mice were randomized into four groups ($n = 10$), undergoing the following treatments: CTR (vehicle) versus MAC (30 mg/kg/oral daily) and/or CIS (8 mg/kg/i.p. once a week) in mono-therapy or in combination therapy. At the end of the treatment (5 weeks), all mice were euthanized and intraperitoneal tumor nodules throughout the peritoneal cavity (including intestine, mesentery, liver and spleen) were counted and analyzed. The number of visible metastases was counted. Values represent the mean ± SD of ten mice in each group from two independent experiments.

**Patient data**. Tumor tissue specimens were provided by Prof. G. Ferrandina from patients admitted to the Gynecologic Oncology Unit, Catholic University of Rome. All patients provided written informed consent for their data to be collected and analyzed for the experimental project; the Catholic University of Rome IRB approved the experimental project as well as the related consent procedures. We also obtained ethic committee approval at the Regina Elena Cancer Institute for the current study. All protocols involving human biospecimens are compliant with all relevant ethical regulations. Tumor tissue specimens were obtained at the time of primary surgery from 30 patients with HG-SOC [stage III]. Because of the unavailability of adequate tissue material from the first cohort of 30 patients to perform immunoblot analysis of ET-1, ET$_A$R, YAP expression, we obtained further HG-SOC specimens at the time of primary surgery. Normal ovarian tissues were obtained from patients who had undergone surgical ovariectomy at the Catholic University of Rome. Biopsies of normal ovarian tissues have been collected tangential to the ovarian surface in order to include also stroma and granulosa cell beneath.Tissues specimens were immediately frozen in liquid nitrogen and then stored at −80 °C until the assay. The linear correlation analysis with $R^2$ and $P$ values between *EDNRA*, *CTGF*, and *CCNA* expression was generated with GraphPad Prism 7 software. IHC analysis of YAP, ET$_A$R, and ET-1 protein expression in HG-SOC tissues and normal tissues was determined from the human protein atlas (www.proteinatlas.org)[70].

**Bioinformatics analysis of TCGA dataset**. Normalized gene expression profiling for HG-SOC patients was obtained from The Cancer Genome Atlas Network (TCGA). Survival analyses were evaluated by Kaplan–Meier method and a log rank test was used to establish the statistical significance of the distance between curves. High and low gene expression values were defined basing on the z-scores of the signals. A multivariate Cox proportional hazard regression model was included to evaluate the impact of clinical variables on the survival curves. A *p*-value less than 0.05 (<0.05) was considered significant. Analyses were performed by MATLAB (The Math Works).

**Statistical analysis**. Except the animal study, each experiment was repeated three times or more. Unless otherwise noted, data are presented as mean ± SD, and statistical analysis was performed using Student's *t*-test to compare two groups of independent samples, and Fisher's exact test for multiple comparisons between groups. For animal study, the time course of tumor growth was compared across the groups using two-way ANOVA, with group and time as variables. All statistical tests were carried out using GraphPad Prism 7.

**Reporting summary**. Further information on research design is available in the Nature Research Reporting Summary linked to this article.

## Data availability
The gene expression data for HG-SOC patients were obtained from Broad Institute TCGA Genome Data Analysis Center: Firehose stddata__2016_01_28 run. Broad Institute of MIT and Harvard. http://gdac.broadinstitute.org/runs/stddata__2016_01_28/data/OV/20160128 or OV https://doi.org/10.7908/C11G0KM9. The clinical data from HG-SOC patients were obtained from the TCGA website at http://www.cbioportal.org[32]. The IHC analysis images were taken from the Human Tissue Atlas database at https://www.proteinatlas.org. All other data generated during this study are available within the article and its supplementary information files or from the corresponding author upon reasonable request. The uncropped scans of the most important blots, can be found in the Supplementary Information. PMOV10 HG-SOC cells will be made available to academic researchers with material transfer agreement.

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

## Acknowledgements

We thank Aldo Lupo for excellent technical assistance, Maria Vincenza Sarcone for secretarial assistance. Funding: This work was supported by Associazione Italiana Ricerca sul Cancro (AIRC) IG18382 (AB) and IG20613 (GB) and Roche per la Ricerca 2016 to A.B.

## Author contributions

P.T. and R.C. performed most of the experiments shown in this work, with the help of V.D.C., L.R., S.D., A.S. G.T., S.B. and G.Bu. provided the sequencing data analysis. E.V.,

G.S., and G.F. provided clinical samples along with clinical annotations from patients. G. B. and A.B. analyzed and discussed the data. A.B. conceived, supervised the project and wrote the paper with input from the other authors. All authors provided comments.

## Additional information

**Competing interests:** G.S. and G.F. have received research funding and consulted for Roche, Tesaro, and AstraZeneca. The remaining authors declare no competing interests.

