## [Peer Review File · Nature Communications]

Reviewers' comments:

Reviewer #1, Expertise: OC
(Remarks to the Author):

NCOMM.182444

Tocci et al

B-arrestin1/YAP/mutant p53 complex orchestrates the endothelin A receptor response in ovarian cancer

This manuscript extends and provides new information concerning the mechanisms by which endothelin (END1; ET-1) activation of the endothelin receptor A promotes ovarian cancer cell proliferation and the induction of specific genes. The authors have previously documented that β -arrestin acts as a dynamic linker of several nuclear transcription factors that impact ovarian cancer growth, including β -catenin, NF- κ B and HIF-1 α . The novelty of the current studies is the presentation of data showing that ET-1 mediated activation of β -arrestin also coordinates the nuclear localization of YAP and interaction with mutant p53, the tethering of these oncogenic factors to the promoters of specific genes (including END1) and their impact on tumor growth in culture and in vivo. In addition, they provide data showing that blocking endothelin receptor activation with macitentan reduces HG-SOC cell proliferation in culture and in vivo and sensitizes cisplatin resistant cells to this drug. The association of ET-1/YAP/p53 with poor patient survival makes targeting this pathway attractive for new therapeutic approaches.

The investigators have utilized an impressive array of approaches (including, Western blots, pull downs, ChIP analyses, proliferation assays, and in vivo xenograft studies) and generated an impressive amount of data to delineate and document the interactions of ET-1, its receptor activation, nuclear localization of YAP and its interactions with mutant p53 on specific target genes. The tremendous amount of data is carefully and clearly presented. The Western blot, pull down and ChIP analyses are of high quality. And the in vivo PDX studies are carefully presented.

Specific Comments:

1. All Figures: The investigators use ovarian cancer cells derived from ovarian cancer patients. PMOV10 cells are from a HG-SOC tumor with a p53 mutant R337T that they identified by sequencing. Other cells are called 2008 (cisplatin sensitive) and 2008-CIS (cisplatin insensitive). The PMOV10 cells and 2008-CIS cells appear to be the most sensitive to ET-1 stimulation. However, because other investigators have not used these cells, and because the p53 mutant R337T present in the PMOV10 cells is not one of the most common in ovarian cancer, it is difficult to generalize. What mutant p53 is present in the 2008 cells +/- Cis? The investigators should document the YAP/mutp53/ β -arrestin complex using one or more of the more common ovarian cancer cell lines, such as OVCAR3 that has a more common p53 mutation (R248Q) or cell lines used in Reference #32 (Di Agostino et al) where YAP has been shown to interact with mutant p53 and NF-Y on genes controlling cell proliferation. Are these cell cycle genes regulated by the YAP/mutp53/ β -arrestin complex in the PMOV10 cells?

2. Based on previous studies by the investigators, are β -catenin, NF κ B or HIF1 α also localized in the YAP/p53mut/ β -arrestin complexes? Some discussion seems warranted.

3. All Figures: For most of the pull-down assays the experiments are terminated at 90 min, apparently, because the nuclear levels of the YAP, β -arrestin, p53 complex appear to decline by

180 min. A time-course pull-down that relates to the induction of the specific genes would seem to be highly relevant; ie at 24 and 72 hours. Even more perplexing is the pull-down of Trio and Rho at 5 min (Figure 3). Some explanation is necessary.

4. Fig 1b. Does total YAP and TAZ levels in cells increase upon ET-1 treatment? From Fig. 1b the cytoplasmic YAP levels remain constant whereas the nuclear YAP and TAZ levels increase upon ET-1 stimulation. It would be good to show a total cell lysate Western to determine if there is an increase in total YAP and TAZ levels as suggested by these results. If so, is this effect at the protein level or mRNA (qPCR)?

5. Based on Fig1b-d the authors conclude in the text (under results section) that they observed a greater accumulation of YAP/TAZ levels in the nuclear compartment of 2008 CIS and PMOV10 cells compared to the 2008 cells. However, none of the data presented in the manuscript allow us to come to this conclusion. In order to make such a conclusion, the authors would need to run samples from these cell lines side by side on the same gel/Western blot. We cannot, compare the signal intensities from different blots and arrive at the conclusion the authors have made as the signal intensities can be a consequence of difference in processing of the blots such as exposure time etc.

6. Fig 1h. Why do we not see much YAP in the cytoplasm? The Western blots in fig1b-d indicate that YAP levels in the cytoplasm remain constant. At least in control and MAC+ET1 treated samples there should be more YAP in the cytoplasm than the nucleus. Have the authors done any quantification of nuclear vs cytoplasmic YAP in the immunofluorescence images presented in Fig 1h? If not, doing so and presenting that data as a bar-graph would help alleviate this concern.

7. Fig. 1i. Have the authors checked the effect of ETBR siRNA on YAP?

8. Fig 2c. The Western blot for YAP on the IP'd β -arrestin for 2008 CIS cells suggests that there is no increase in YAP levels upon ET-1 stimulation as one would expect. Even if there is an increase, the effect is very modest unlike for the 2008 cells. This result does not agree with the authors conclusion based on Fig 1 the effect of ET-1 on nuclear localization of YAP is more robust in 2008 CIS cells compared to 2008 cells. How do the authors explain this? Do they suspect dissociation of YAP and β -arrestin after entering the nucleus in 2008 CIS cells resulting in less than expected YAP pull down using β -arrestin antibody? If so, why is that not the case with 2008 cells? Also, why are we seeing such high levels of YAP being pulled down with β -arrestin from the nucleus even without ET-1 stimulation in 2008 CIS cells? Fig 1c shows hardly any YAP being present without ET-1 stimulation in these cells.

9. In Fig 2e, were the cells treated with ET-1? If not, have the authors tested that to rule out if that would impact the IP with the mutant?

10. Fig 2f. Why does ET-1 not have an effect of YAP and TAZ nuclear levels in cells rescued with β -arrestin-FLAG? If anything, the levels of YAP seem to be decreasing in nucleus upon ET-1 stimulation, which is opposite to the expectation based on previous data. Also, the authors need to show total extract or at least the cytosol Westerns to confirm that β -arresting knockdown has worked as expected.

11. Fig 3h. What is the effect of the actin disrupting agents on LATS1 levels and/or phosphorylation status? Showing these data are important to make the connection the authors make in the manuscript between actin, LATS1 and YAP.

12. Figure 4: Do cells expressing other p53 mutants also co-localize with β -arrestin and YAP in the ChIP analyses? Is there a control promoter that does not bind YAP be used in the ChIP assays? This should be included. Do promoters of the cell proliferation genes that bind YAP/p53/NF-Y also bind β -arrestin?

13. Figure 6: The figure legend does not make clear that the PDX tumors are from cells injected subcutaneously and that the small nodules are derived from tumor cells injected ip. Does macitentan inhibit tumor growth of other more commonly used ovarian cancer cell lines? The effects of macitentan are impressive, but how general is this phenomenon? Some attention should be given to this. What are the levels of β -arrestin and mutp53 in the tumors and the nodules tissues? These should be included in the Western blots because they are in the ChIP assays.

Reviewer #2, Expertise: Endothelin (Remarks to the Author):

GENERAL COMMENTS TO AUTHORS:

Thank you for asking me to review the manuscript by Tocci et al, from Prof Bagnato's team, entitled " β -arrestin1/YAP/mutant p53 complex orchestrates the endothelin A receptor response in ovarian cancer".

The paper investigates mechanisms to explain the limited clinical response observed in high-grade serous ovarian cancer (HG-SOC), with high frequency of TP53 mutations (mutp53).

The work describes how upon activation of Endothelin receptors (ET-1R), β -arrestin1 (β -arr1), which controls G-protein-coupled receptor signaling (including ET-1R), interacts with YAP, triggering its cytoplasmic-nuclear shuttling, recruiting mutp53 to the YAP-TEAD transcriptional complex. In parallel, β -arr1 mediated the ET-1R-induced Trio/RhoA-dependent YAP nuclear accumulation. And that the overall result was the activation of YAP/TEAD target genes, including EDN1 that ensures persistent signals sustaining aggressive traits.

The work was carried out in patient derived cancer cells and platinum resistant cancer cells. In vivo work in patient-derived xenografts showed synergy between a dual ET-1R antagonist (macitentan) and cisplatin. And that the anti-cancer effects were mediated via shutting-down the β -arr1-mediated YAP/mutp53/TEAD transcriptional program. In parallel, ETAR/ β -arr1/YAP gene signature correlated with a worst prognosis in HG-SOC.

The authors propose that these preclinical results champion repurposing of macitentan for combinatorial treatment of HG-SOC.

GENERAL SCIENTIFIC COMMENTS:

The work is robust, extensive and addresses the specific scientific questions coherently. The use of models from in vitro to in vivo and importantly using patient derived cells is appropriate and well thought out. A number of methodologies are demanding and models range from the molecular to the cellular to whole tissue/organism. The conclusions are well supported and well thought out.

SPECIFIC COMMENTS:

Under Summary:

The phrase: "oncogenic pathway network" is rather vague. Please change to provide more focus to the whole first sentence. Minor suggestion: remove commas in this sentence.

Under introduction:

Line 4: "Considering the inefficacy of chemotherapy to greatly enhance HG-SOC patient outcomes...". Suggestion: "considering the low efficacy..."

Under Results, Page 5:

"In all cells used ETAR and ETBR were expressed and PMOV10 and 2008 CIS cells expressed higher levels of ETAR compared to sensitive 2008 cells (Fig. 1a). In these cells ET-1, in a time-

dependent manner, promoted the reduction of pYAP (S127) and pTAZ (S89) in the cytoplasm paralleled with YAP/TAZ nuclear accumulation (Fig. 1b-d)." Comment: First sentence construction is awkward, please re-write. Second sentence: "in these cells", it is not clear whether these cells refer to "all cells" or just PMOV and 2008CIS. Please clarify.

Relevant figures: The authors put a lot of effort into their figures and they mostly optimally presented. Have they considered whether additional tables could be incorporated into the complex figures (especially 2, 4) to define / state significant differences between key groups (e.g., for 2h). This would remove the "busy" appearance of multiple stars on top of histogram columns.

Under Results, Page 12:

Next, we evaluated the response of PMOV10 cells to clinically relevant doses of cisplatin and we found that these cells were poor responsive to cisplatin-induced apoptosis even at the highest concentrations used (Fig. 5c). Comment: correct "poor responsive". Give reference for clinically relevant doses of cisplatin.

Under Methods:

Stat analysis was performed using Students t-test and Fisher's exact test to compare in vitro experiments. Comment: why was ANOVA not used in some in vitro experiments?

Marilena Loizidou

Reviewer #3, Expertise: YAP, cancer
(Remarks to the Author):

Tocci et al present a compelling paper with regards to the role of endothelin receptor and Yap/Taz cooperating in ovarian cancer to result in cisplatin-based chemotherapy resistance. Potentially, it could have a potent impact in the field of ovarian oncology because of their demonstration of a FDA approved drug, macitentan to reduce YAP activity and improve cisplatin sensitivity.

My only major critique is it is unclear from the presented patient data (Figure 7), the frequency that ET-1 inhibition would impact sensitivity to cisplatin-based therapy through the Hippo pathway. They present data Figure 7a-c from ~20 patients regarding a positive mRNA relationship between EDNRNA, YAP, ANKRD1, and Arrestin. The correlation coefficients are rather modest and more importantly, their model suggests that high EDNRNA levels results in changes in Yap level at the post-translational level. Low Yap transcription may still result in high Yap levels because signaling through the endothelin receptor promotes Yap stability and nuclear localization. ANKRD1, a proposed proxy of Hippo pathway activity (Figure 7b) generally appears to be expressed at a very low level suggesting that Yap is not very active. This would not support that their model is generally active in many cases of ovarian cancer.

If possible, the authors should present immunoblotting of Yap, ET-1, endothelin Receptor from a cohort of cases against "normal" ovarian tissue to give readers a better assessment of the frequency that their proposed model may be valid in cases of ovarian cancer.

A minor, but relatively important aspect to address would also entail the readability of the introduction. It was poorly structured with numerous grammatically errors. The results section and discussion were written with much more care and thoughtfulness, but the introduction was difficult to digest. It should be significantly rewritten to better introduce the topic and potential importance of their findings.

Minor:

Figure 2 - 4: Immunoblots are labeled on the side-bar with IP when they should be labeled IB for immunoblotting.

We thank the reviewers for the helpful and insightful comments and suggestions, all of which have greatly improved our manuscript. We have revised the manuscript addressing mostly the reviewer's comments and modified it accordingly.

Following is our point-by-point response to the reviewer's comments.

Reviewer #1 Expertise OC:

Comment 1. All Figures: The investigators use ovarian cancer cells derived from ovarian cancer patients. PMOV10 cells are from a HG-SOC tumor with a p53 mutant R337T that they identified by sequencing. Other cells are called 2008 (cisplatin sensitive) and 2008-CIS (cisplatin insensitive). The PMOV10 cells and 2008-CIS cells appear to be the most sensitive to ET-1 stimulation. However, because other investigators have not used these cells, and because the p53 mutant R337T present in the PMOV10 cells is not one of the most common in ovarian cancer, it is difficult to generalize. What mutant p53 is present in the 2008 cells +/- Cis? The investigators should document the YAP/mutp53/ β -arrestin complex using one or more of the more common ovarian cancer cell lines, such as OVCAR3 that has a more common p53 mutation (R248Q) or cell lines used in Reference #32 (Di Agostino et al.) where YAP has been shown to interact with mutant p53 and NF-Y on genes controlling cell proliferation. Are these cell cycle genes regulated by the YAP/mutp53/ β -arrestin complex in the PMOV10 cells?

Response: We thank the reviewer for these thoughtful and constructive comments pointing out to use more common ovarian cancer cell lines carrying more common TP53 mutations. This is to demonstrate further that ET-1R signaling promotes the formation of β -arrestin/YAP/mutp53 complex. To pursue it: (A) we performed almost all the in vitro and in vivo experiments (new FIG.1, 2, 4, and 6), using OVCAR-3, a more common ovarian cancer cell line that carries a hot spot missense TP53 mutation (R248Q) (New Fig.1). We confirmed the presence of the trimeric complex β -arr1/YAP/mutp53, by co-IP and ChIP analyses, as also documented in patient-derived PMOV10 cells. Of note, in OVCAR-3 cells mut-p53 depletion hampered YAP and β -arr1 interaction, and similarly YAP depletion impaired β -arr1 and mutp53 binding, suggesting that β -arr1 may provide a nuclear anchor for mutp53 and YAP proteins to activate the expression of downstream target genes (New Fig.4b, d, f); (B) we have also demonstrated that ET-1 promoted YAP/TAZ nuclear accumulation leading to the formation of the trimeric complex β -arr1/YAP/mutp53 in a breast cancer cell line (MDA-MB-468) carrying one of the most common p53 mutants (R273H) (New Fig.1d) and expressing both ET_AR and ET_BR (New Supplementary Fig.1c). In this cellular context it has been previously demonstrated that YAP and mutp53 proteins physically interact with NF-Y controlling transcriptionally cell cycle regulated genes and thereby cell proliferation (Di Agostino S. et al. EMBO Rep. 2016); (C) we also found that the β -arrestin-1/YAP/mutp53 complex regulated transcriptionally and aberrantly the expression of cell cycle genes (CCNA and CDK1) (Fig.4n) in PMOV10 cells. Altogether, these findings originally demonstrate that β -arrestin-1, mutp53 and YAP can physically associate in the nucleus and drive pro-proliferative gene expression, such as cyclin A (CCNA). This strongly suggests that β -arrestin-1/mutp53/YAP might represent a critical oncogenic node activating aberrantly a mutp53-dependent transcriptional signature that can be interfered by the ET-1R antagonist, macitentan. Therefore in the revised version we provide additional data on the mechanisms and therapeutic relevance of blocking ET-1R in HGSOC with common TP53 mutations (Fig.6). Collectively, these new data provide clear mechanistic insights on the role of YAP/mutp53/ β -arrestin complex upon ET-1 stimulation in ovarian cancer (OVCAR3) and breast cancer cell lines (MDA-MB-468), both carrying common TP53 mutations. In light of these results in the revised version of the manuscript we left out the data on cell lines that do not closely represent HG-SOC, 2008 and 2008 CIS, due to both their uncommon use as ovarian cancer preclinical setting and type of TP53 mutation.

Comment 2. Based on previous studies by the investigators, are β -catenin, NF κ B or HIF1 α also localized in the YAP/p53mut/ β -arrestin complexes? Some discussion seems warranted.

Response: We agree with reviewer that assessing the potential presence of other transcriptional factors, such as HIF1 α , within the β -arrestin-1/YAP/mutp53 complexes is an important question that we aim to address in a future work. In the revised Discussion of the manuscript we added: “Moreover, our findings point toward a role for β -arr1/YAP/mutp53 as a major nexus that integrates and decodes different stimuli into precise transcriptional programs in HG-SOC upon ETAR activation. The data presented here describe an unpredicted mechanism by which ET-1R/ β -arr1, engages the oncogenic cross-talk between mutp53 and YAP, amplifying aberrantly YAP activity in HG-SOC. It might also be that β -arr1 can anchor different transcription factors, as we report for NFY, to expand its own agenda providing a selective advantage to HGSOC cells. Hence, our study provides novel mechanistic insights by which β -arr1-mutp53-YAP complex represents the initial scaffold on which transcriptional regulatory networks could be built to regulate different biological outcomes. It has been increasingly clear (Di Agostino S. et al., Embo Rep. 2006; Weisz L. et al. Cancer Res. 2007, Cooks T. et al. Cancer Cell 2013; Amelio I. et al. Proc Natl Acad Sci U S A. 2018; Verduci L. et al Genome Biol., 2017) that the interactions with active transcription factors, tethering mutp53 to the DNA, can mediate and broaden the ability of gain of function mutp53 proteins to regulate gene expression. Further studies should attempt to clarify whether ET-1, through distinct β -arr1-complexes with YAP and mutp53 can recruit or interact with factors, such as HIF-1 α , β -catenin, or NF κ B, to orchestrate interconnected pathway network regulating tumor growth and progression in HG-SOC, and in other types of cancer harbouring gain of function TP53 mutations”.

[REDACTED]

Comment 3. All Figures: For most of the pull-down assays the experiments are terminated at 90 min, apparently, because the nuclear levels of the YAP, β -arrestin, p53 complex appear to decline by 180 min. A time-course pull-down that relates to the induction of the specific genes would seem to be highly relevant; ie at 24 and 72 hours. Even more perplexing is the pull-down of Trio and Rho at 5 min (Figure 3). Some explanation is necessary.

Response: We thank the reviewer for this comment. As requested, we repeated the IP at longer times, showing that even at 12-24h the β -arrestin/YAP/p53 complex is still present (new Supplementary 4b) pairing the induction of target genes. This confirms that β -arr1-dependent signaling can engender highly characteristic transcriptomic phenotypes and generate long-lasting effects through the formation of a multi-protein transcription competent complex. Similarly, we repeated the kinetics of IP of Trio and Rho at different time-points (new Fig.3a), confirming the presence of the interaction at longer times.

Comment 4. Fig 1b. Does total YAP and TAZ levels in cells increase upon ET-1 treatment? From Fig. 1b the cytoplasmic YAP levels remain constant whereas the nuclear YAP and TAZ levels increase upon ET-1 stimulation. It would be good to show a total cell lysate Western to determine if there is an increase in total YAP and TAZ levels as suggested by these results. If so, is this effect at the protein level or mRNA (qPCR)?

Response: As suggested by the reviewer, we evaluated the effect of ET-1 stimulation on the expression of total YAP and TAZ levels. As shown in the new Fig. 1g-j, in both PMOV10 and OVCAR-3 cells, ET-1 upregulated YAP and TAZ at both mRNA and protein levels, suggesting that ET-1 induces transcriptionally YAP/TAZ expression.

Comment 5. Based on Fig1b-d the authors conclude in the text (under results section) that they observed a greater accumulation of YAP/TAZ levels in the nuclear compartment of 2008 CIS and PMOV10 cells compared to the 2008 cells. However, none of the data presented in the manuscript allow us to come to this conclusion. In order to make such a conclusion, the authors would need to run samples from these cell lines side by side on the same gel/Western blot. We cannot, compare the signal intensities from different blots and arrive at the conclusion the authors have made as the signal intensities can be a consequence of difference in processing of the blots such as exposure time etc.

Response: As previously explained, we removed the data on 2008 and 2008 CIS cells, focusing on patient-derived HGSOC cells (PMOV10) and more common used cell lines carrying common TP53 mutations (OVCAR3 and MDA-MB-468) (new Fig. 1)

Comment 6. Fig 1h. Why do we not see much YAP in the cytoplasm? The Western blots in fig1b-d indicate that YAP levels in the cytoplasm remain constant. At least in control and MAC+ET1 treated samples there should be more YAP in the cytoplasm than the nucleus. Have the authors done any quantification of nuclear vs cytoplasmic YAP in the immunofluorescence images presented in Fig 1h? If not, doing so and presenting that data as a bar-graph would help alleviate this concern.

Response: As suggested by the reviewer, we performed the IF in OVCAR3 cells and quantified nuclear vs cytoplasmic YAP in the immunofluorescence images presented in the new Fig.1f. We found that in the control and MAC+ET-1, YAP is predominantly stained in the cytoplasm while in ET-1-treated cells YAP is enriched in the nucleus, as now presented in the new bar-graph (New Fig.1F).

Comment 7. Fig. 1i. Have the authors checked the effect of ETBR siRNA on YAP?

Response: As suggested by the reviewer, we analyzed the effect of ETBR siRNA-mediated depletion, confirming the data obtained with BQ788, an ETBR selective antagonist (New Suppl. Fig.1f). Thus, ETBR siRNA did not decrease YAP nuclear accumulation, demonstrating that ET-1 acts mainly through ETAR to control the nuclear trafficking of YAP in HG-SOC cells.

Comment 8. Fig 2c. The Western blot for YAP on the IP'd β -arrestin for 2008 CIS cells suggests that there is no increase in YAP levels upon ET-1 stimulation as one would expect. Even if there is an increase, the effect is very modest unlike for the 2008 cells. This result does not agree with the authors conclusion based on Fig 1 the effect of ET-1 on nuclear localization of YAP is more robust in 2008 CIS cells compared to 2008 cells. How do the authors explain this? Do they suspect dissociation of YAP and β -arrestin after entering the nucleus in 2008 CIS cells resulting in less than expected YAP pull down using β -arrestin antibody? If so, why is

that not the case with 2008 cells? Also, why are we seeing such high levels of YAP being pulled down with β -arrestin from the nucleus even without ET-1 stimulation in 2008 CIS cells? Fig 1c shows hardly any YAP being present without ET-1 stimulation in these cells.

Response: As previously explained, we removed the data on 2008 and 2008 CIS cells, that do not closely represent HG-SOC, focusing on OVCAR3 and PMOV10 cells in which increased YAP levels upon ET-1 stimulation were clearly showed in the Western blot for YAP on the IP'd β -arrestin (new Fig. 2 a-c and new Fig.4 a-g).

Comment 9. In Fig 2e, were the cells treated with ET-1? If not, have the authors tested that to rule out if that would impact the IP with the mutant?

Response: As suggested by the reviewer, we repeated the experiment in the absence and in the presence of ET-1 (new Fig. 2c). We found that the β -arr1 deletion mutant, lacking of a region containing a sequence required for its nuclear localization, was unable to co-IP with YAP upon ET-1 stimulation, indicating that β -arr1 functions as a co-pilot of ET_AR in regulating YAP nuclear localization.

Comment 10. Fig 2f. Why does ET-1 not have an effect of YAP and TAZ nuclear levels in cells rescued with β -arrestin-FLAG? If anything, the levels of YAP seem to be decreasing in nucleus upon ET-1 stimulation, which is opposite to the expectation based on previous data. Also, the authors need to show total extract or at least the cytosol Westerns to confirm that β -arrestin knockdown has worked as expected.

Response: We apologize for the lack of clarity regarding the YAP and TAZ nuclear levels rescued with ectopic expression of β -arr1-FLAG reported in Fig. 2F. To address this point, following optimization of our reaction conditions, we have repeated our experiments. As reported in the new Fig.2d, the new blots showed that the cells transfected with β -arr1-Q394L, in which the nuclear export signal was introduced by a single point (Q394L) mutation, were characterized by a reduction of YAP/TAZ nuclear accumulation. This effect was rescued by the re-expression of β -arr1 and ET-1 stimulation further increased YAP/TAZ accumulation in cells rescued with β -arr1-FLAG. The silencing of β -arr1 on the total extracts of PMOV10 and OVCAR3 cells is now shown in new Suppl. Fig.2a,b,g.

Comment 11. Fig 3h. What is the effect of the actin disrupting agents on LATS1 levels and/or phosphorylation status? Showing these data are important to make the connection the authors make in the manuscript between actin, LATS1 and YAP.

Response: As suggested by the reviewer, we performed the experiment on LATS1 phosphorylation status in the absence and in the presence of actin disrupting agents (new Fig. 3h). We found that the regulation of LATS activity requires a functional actin cytoskeleton that contributes to ET-1R/ β -arr1/RhoA-mediated activation of YAP signaling.

Comment 12. Figure 4: Do cells expressing other p53 mutants also co-localize with β -arrestin and YAP in the ChIP analyses? Is there a control promoter that does not bind YAP be used in the ChIP assays? This should be included. Do promoters of the cell proliferation genes that bind YAP/p53/NF-Y also bind β -arrestin

Response: As requested by the reviewer, we performed new ChIP analyses and confirmed the presence of the trimeric complex β -arr1/YAP/mutp53 in OVCAR3 (mutp53R248Q) cells (Fig.4k) and in MDA-MB-468 cells (mutp53R273H) (Suppl. Fig. 4j) onto target gene promoters. Moreover, a control promoter (HBB, hemoglobin), which does not contain YAP/TEAD binding sites, has been included in the ChIP assays (new Fig.4k and Suppl. Fig.4j). Of note, promoters of the cell proliferation (CCNA and CDK1) onto which YAP/p53/NF-Y complex was recruited (Di Agostino et al. EMBO Rep. 2016) also bind β -arrestin, as shown in the new Suppl. Fig.4j. We thank the reviewer for this insightful comment pointing to strengthen the demonstration that β -arrestin-1-mutp53-YAP represents the initial regulatory tethering hub on which transcriptional regulatory complexes could be built, as we report for NFY (new Suppl. Fig. 4j). This allows eliciting different oncogenic effects in HGSOC, and in

other types of cancer, such as breast cancer, carrying gain of function p53 mutations, as now discussed in the manuscript and commented in the response 2.

Comment 13. Figure 6: The figure legend does not make clear that the PDX tumors are from cells injected subcutaneously and that the small nodules are derived from tumor cells injected ip. Does macitentan inhibit tumor growth of other more commonly used ovarian cancer cell lines? The effects of macitentan are impressive, but how general is this phenomenon? Some attention should be given to this. What are the levels of β -arrestin and mutp53 in the tumors and the nodules tissues? These should be included in the Western blots because they are in the ChiP assays.

Response: We apologize for the lack of clarity regarding the figure legend of Fig. 6 that in its revised version explains more clearly that the representative PDX tumors (at right of the fig.6a-c) are from patient-derived PMOV10 cells transplanted subcutaneously, whereas the tumor nodules derived from PMOV10 cells injected ip are shown in Fig. 6d,e. We previously demonstrated the inhibitory effect of macitentan on the tumor growth of other more commonly used ovarian cancer cell lines as A2780, HEY and SKOV3 cells (Rosanò L et al. Cancer Res. 2014, Cianfrocca R. et al. Oncotarget 2016, Semprucci E. et al. Oncogene 2015, Di Modugno F. et al. PNAS 2018). Of relevance, new results added in the revised version of the manuscript demonstrate for the first time that macitentan can reduce the number of tumor nodules in HG-SOC OVCAR-3 xenografts (Fig.6f), highlighting the therapeutic relevance of blocking ET-1R by macitentan in HGSOC with hot spot missense TP53 mutations. The effect was more significant in those mice treated with MAC+CIS (Fig. 6f) and was paralleled by significantly increased pYAP protein expression levels, as evidenced by immunoblotting analysis of metastatic nodules of OVCAR-3 xenografts (new Fig 6g).

The therapeutic benefit of the FDA and EMA approved for pulmonary arterial hypertension dual ET-1R antagonist macitentan is to target not only HG-SOC cells expressing ET_AR, but also to interfere with TME elements, such as vascular, lymphatic, fibroblast and inflammatory cells, which mainly express ET_BR (Rosanò L, Spinella F, Bagnato A. Nat. Rev. Cancer 2013) representing therefore a therapeutic strategy which may be used to design effective combinatorial targeted therapies. To appreciate how the effect of macitentan could be generalized, we should take in account that ET-1R blockade by macitentan, sensitizes tumor cells to different cytotoxic and molecular agents in various preclinical tumor models, including ovarian, colorectal cancer, glioblastoma, multiple myeloma, breast and lung brain metastasis (Kim SJ, et al. Neoplasia. 2011; Rosanò L et al. Cancer Res. 2014; Sestito R et al. Life Sci. 2016; Cianfrocca R et al. Cell Death Differ. 2017; Kim SJ, et al. Clin Cancer Res. 2015; Russignan A. et al. Br J Haematol. 2019; Lee HJ et al. Neuro Oncol. 2016, Askoxylakis V et al. NPJ Breast Cancer. 2019), as now reported in the Introduction, and therefore may represent a new therapeutic option for cancer patients that can impair YAP/mutp53 transcriptional machinery, as now reported in discussion.

The levels of β -arrestin and mutp53 in the immunoblotting of tumors and nodules tissues have been now included in New fig. 6g and new Suppl. Fig.6f,g.

Reviewer #2 Expertise Endothelin

Comment: Under Summary: The phrase: “oncogenic pathway network” is rather vague. Please change to provide more focus to the whole first sentence. Minor suggestion: remove commas in this sentence.

Response: Thanks for pointing this out. We changed the sentence with “The limited clinical response observed in high-grade serous ovarian cancer (HG-SOC) with high frequency of TP53 mutations (mutp53) might be related to the mutp53-driven oncogenic pathway network.”, in the revised summary.

Comment: Under introduction: Line 4: “Considering the inefficacy of chemotherapy to greatly enhance HG-SOC patient outcomes...”. Suggestion: “considering the low efficacy...”

Response: We fully modified this sentence under Introduction.

Comment: Under Results, Page 5: “In all cells used ETAR and ETBR were expressed and PMOV10 and 2008 CIS cells expressed higher levels of ETAR compared to sensitive 2008 cells (Fig. 1a). In these cells ET-1, in a time-dependent manner, promoted the reduction of pYAP (S127) and pTAZ (S89) in the cytoplasm paralleled with YAP/TAZ nuclear accumulation (Fig. 1b-d).” Comment: First sentence construction is awkward, please re-write. Second sentence: “in these cells”, it is not clear whether these cells refer to “all cells” or just PMOV and 2008CIS. Please clarify.

Response: We changed both sentences in the revised version of the manuscript. As we previously explained, we added two tumor cell lines (OVCAR-3, ovarian and MDA-MB-468, breast) bearing common TP53 mutations and we left out the data on 2008 and 2008 CIS cells, due to both their uncommon use as ovarian cancer preclinical setting and type of TP53 mutations. In the first sentence: “PMOV10, OVCAR-3 cells expressed both ETAR and ETBR (Fig. 1a)”. In the second sentence, we clarify: “ In both cells.....” and now we added “Similarly, in a breast cancer cell line (MDA-MB-468) carrying a common TP53 mutation (R273H) and expressing ET_AR and ET_BR, ET-1 induced YAP/TAZ dephosphorylation and nuclear accumulation”.

Comment: Relevant figures: The authors put a lot of effort into their figures and they mostly optimally presented. Have they considered whether additional tables could be incorporated into the complex figures (especially 2, 4) to define / state significant differences between key groups (e.g., for 2h). This would remove the “busy” appearance of multiple stars on top of histogram columns.

Response: Thanks for pointing this out. We try to optimally present the new data in the relevant figures, especially new Fig. 2 and 4 and new Supplementary Fig.4, making every attempt to appropriately modify the figures, following the journal style.

Comment: Under Results, Page 12: Next, we evaluated the response of PMOV10 cells to clinically relevant doses of cisplatin and we found that these cells were poor responsive to cisplatin-induced apoptosis even at the highest concentrations used (Fig. 5c). Comment: correct “poor responsive”. Give reference for clinically relevant doses of cisplatin.

Response: To clarify the text, we rephrased the sentence: “Next, we evaluated the response of PMOV10 cells to different doses of cisplatin and we found that the combination of cisplatin with macitentan, rendered PMOV10 cells more prone to cisplatin-induced cell death (Fig. 5c)”. Moreover we added the reference for the different doses of cisplatin (Zhang, B. et al. *Oncology Reports*, 29, 1371-1378, 2013).

Comment: Under Methods: Stat analysis was performed using. Comment: why was ANOVA not used in some in vitro experiments?

Response: In our in vivo experiments, the time course of tumor growth was compared across the groups using two-way ANOVA, using two independent variables (group and time).

We performed Students t-test and Fisher’s exact test in in vitro experiments to clearly present the data, especially in Fig. 2 and 4, and avoid the “busy” appearance of multiple stars and connecting lines on top of histogram columns compared to multiple conditions (i.e. Ctr or ET-1 stimulated cells).

Reviewer #3, Expertise: YAP, cancer

Comment: My only major critique is it is unclear from the presented patient data (Figure 7), the frequency that ET-1 inhibition would impact sensitivity to cisplatin-based therapy through the Hippo pathway. They present data Figure 7a-c from ~20 patients regarding a positive mRNA relationship between EDNRB, YAP, ANKRD1, and Arrestin. The correlation coefficients are rather modest and more importantly, their model suggests that high EDNRB

levels results in changes in Yap level at the post-translational level. Low Yap transcription may still result in high Yap levels because signaling through the endothelin receptor promotes Yap stability and nuclear localization. ANKRD1, a proposed proxy of Hippo pathway activity (Figure 7b) generally appears to be expressed at a very low level suggesting that Yap is not very active. This would not support that their model is generally active in many cases of ovarian cancer.

Response: We thank the reviewer for this thoughtful and constructive comment pointing to strengthen the clinical relevance of the ET-1R signaling and YAP in HG-SOC patients characterized by high TP53 mutation rate. From TCGA analysis, it has been reported that different types of cancers have distinctive patterns of Hippo-p53 axis activation. Interestingly in tumors in which there is a strong pressure to mutate TP53, CCNA, along with CTGF, is expressed at higher levels, suggesting the existence of a transcriptionally active mutp53-YAP complex (Furth N., Aylon Y., Oren M., Cell Death Differ. 2018). On the basis of this consideration, we evaluated the correlation in HG-SOC patients of CCNA, CTGF and EDNRA and we found a more robust and statistically significant association between CCNA, CTGF and EDNRA in our cohort of 30 patients (new Fig.7a,b). These distinctive patterns of YAP-mutp53 axis strengthen the clinical relevance of the cooperative signaling between ETAR and YAP in HG-SOC.

Comment: If possible, the authors should present immunoblotting of Yap, ET-1, endothelin Receptor from a cohort of cases against "normal" ovarian tissue to give readers a better assessment of the frequency that their proposed model may be valid in cases of ovarian cancer.

Response: As suggested by the reviewer, we analyzed the relationship between YAP, ET_AR and ET-1 protein expression in a group of 21 HG-SOC patient tissues and in non-tumoral ovarian tissues (n=6). We found that the expression YAP, ET_AR and ET-1 in normal ovarian tissues derived from ovariectomy were lower and more homogeneous compared to those from HG-SOC tissues (Fig. 7c). These findings suggest that YAP overexpression is associated with ET_AR/ET-1 axis in a high percentage of HG-SOC compared to normal ovarian tissues, reflecting the close network between the ETAR and YAP pathways in HG-SOC.

Comment: A minor, but relatively important aspect to address would also entail the readability of the introduction. It was poorly structured with numerous grammatically errors. The results section and discussion were written with much more care and thoughtfulness, but the introduction was difficult to digest. It should be significantly rewritten to better introduce the topic and potential importance of their findings.

Response: Thanks for pointing this out. We re-write the introduction to better introduce the topic and potential importance of their findings. The entire manuscript has been proofread and edited for grammatical and spelling mistakes.

Minor:

Figure 2 - 4: Immunoblots are labeled on the side-bar with IP when they should be labeled IB for immunoblotting.

Response: Thanks for pointing this out. We added these changes in the IP of Fig.2-4.

Best regards,

Anna Bagnato

REVIEWERS' COMMENTS:

Reviewer #1 (Remarks to the Author):

β -arrestin1/YAP/mutant p53 complex orchestrates the endothelin A receptor
TocciBagnato

In this revised manuscript, the authors have carefully responded to all of the reviewers' comments and suggestions to improve the scientific basis of endothelin A receptor signaling in high-grade serous ovarian cancer.

This is a very nice contribution.

Minor comments:

Line 96: "with to" should be either with or to.

Line 122: A space is needed between used and OVCAR3

Line 364-365: Are the differences related to cell types in the ovaries, ie. Granulosa cells, stroma versus tumor. Are there any IHC data on this in the Human Tissue Atlas? This would be nice to include if it is available.

Line 410: increasing clear should be increasingly clear

Reviewer #2 (Remarks to the Author):

The changes made address fully the original comments, with an excellent final paper.

Marilena Loizidou

Reviewer #3 (Remarks to the Author):

Tocci et al submit a compelling paper regarding an Endothelin A receptor - Yap - b-arrestin - mut p53 axis which appears important in high grade serous ovarian cancer. Such cancer confer high mortality and have limited therapeutic options. This study highlights the use of an FDA approved drug (macitentan), an inhibitor of Endothelin A Receptor to disrupt the cooperative the Yap - b-arrestin - mut p53 signaling and potentiating standard cisplatin therapy.

My previous review only concerned Figure 7 and the relevance from external data that Hippo signaling actually plays a role in this form of ovarian cancer. The authors have reanalyzed the data focusing on a subset of ovarian cancer and have found correlation coefficients and significance that would strongly support the investigation of ET1RA inhibition as a means to disrupt Hippo/Yap signaling and potentiate standard chemotherapy.

My co-reviewers suggested using other ovarian cancer cell lines which have more common p53 mutations and well characterized cell lines to ensure the results Tocci et al discovered are not limited to a minor subset of ovarian cancer. The previous results of such a Yap-b arrestin-p53 complex seems to be confirmed and robust in other conventional cell lines.

As a result, I endorse strong consideration of publication of this manuscript, because they demonstrate a robust and novel oncogenic mechanism with a drug that should be available to patients because of its use in other fields. The paper is likely to have high impact in the ovarian cancer field where few clinical options exist, and this work may translate into other fields as well. I would suggest a few minor revisions for the purpose of clarity.

Minor:

- Figure 2B-D: Labels across the top of Figure 2B-D are too tight making it difficult to interpret the data.
- Figure 5e: Please quantify these blots. Differences are quite modest and not easily visualized. Represent as an associated graph or in the supplemental data.
- Minor grammatical errors throughout the manuscript. Please edit for grammar.

Following is our point-by-point response to the reviewer's comments.

Reviewer #1 Expertise OC:

Comment: In this revised manuscript, the authors have carefully responded to all of the reviewers' comments and suggestions to improve the scientific basis of endothelin A receptor signaling in high-grade serous ovarian cancer. This is a very nice contribution.

Response: Thanks for your previous great suggestions. It helps us a lot to improve our work.

Comment: Line 96: "with to" should be either with or to.

Line 122: A space is needed between used and OVCAR3

Line 410: increasing clear should be increasingly clear

Response: Thanks very much. We have corrected.

Comment: Line 364-365: Are the differences related to cell types in the ovaries, i.e. Granulosa cells, stroma versus tumor. Are there any IHC data on this in the Human Tissue Atlas? This would be nice to include if it is available.

Response: Thanks for this interesting question. In this study, biopsies of normal ovarian tissue have been collected tangential to the ovarian surface in order to provide a good balance between ovarian surface epithelium and stromal tissue beneath, as now specified in Methods section. Therefore, it is possible that granulosa cells as well as fibroblasts could be present in the cell lysates that we analyzed by western blot for the protein expression. We also thank for the suggestion to use IHC data in the Human Tissue Atlas. As suggested, we now included in Supplementary Fig. 7e the IHC analysis of YAP, ETAR and ET-1 protein expression in clinical specimens from the human protein atlas. YAP, ETAR and ET-1 displayed strong expression in HGSOC, compared to ovarian tissues, as now added in the Results section.

Reviewer #2 Expertise Endothelin

Comment: The changes made address fully the original comments, with an excellent final paper

Response: We thank the reviewer for their kind words.

Reviewer #3, Expertise: YAP, cancer

Comment: Tocci et al submit a compelling paper regarding an Endothelin A receptor - Yap - b-arrestin - mut p53 axis which appears important in high grade serous ovarian cancer. Such cancers confer high mortality and have limited therapeutic options. This study highlights the use of an FDA approved drug (macitentan), an inhibitor of Endothelin A Receptor to disrupt the cooperative the Yap - b-arrestin - mut p53 signaling and potentiating standard cisplatin therapy. My previous review only concerned Figure 7 and the relevance from external data that Hippo signaling actually plays a role in this form of ovarian cancer. The authors have reanalyzed the data focusing on a subset of ovarian cancer and have found correlation coefficients and significance that would strongly support the investigation of ET1RA inhibition as a means to disrupt Hippo/Yap signaling and potentiate standard chemotherapy. My co-reviewers suggested using other ovarian cancer cell lines which have more common p53 mutations and well characterized cell lines to ensure the results Tocci et al discovered are not limited to a minor subset of ovarian cancer. The previous results of such a Yap-b arrestin-p53 complex seems to be confirmed and robust in other conventional cell lines. As a result, I endorse strong consideration of publication of this manuscript, because they demonstrate a robust and novel oncogenic mechanism with a drug that should be available to patients because of its use in other fields. The paper is likely to have high impact in the ovarian cancer field where few clinical options exist, and this work may translate into other fields as well.

I would suggest a few minor revisions for the purpose of clarity.

Response: Thank you for your very helpful suggestions. It definitely improved our manuscript.

Comment: Figure 2B-D: Labels across the top of Figure 2B-D are too tight making it difficult to interpret the data.

Response: Thanks very much. We have corrected all the figures.

Comment: Figure 5e: Please quantify these blots. Differences are quite modest and not easily visualized. Represent as an associated graph or in the supplemental data.

Response: We appreciate this comment and Figure 5e indeed show quite modest differences and not easily visualized. Therefore, since the Figure 5f and g included the data showed in the old Fig. 5d and e, demonstrating also that enhanced levels of cleaved PARP, caused by macitentan and/or CIS treatments associated to β -arr1 or YAP depletion, was completely recovered by ectopic expression of β -arr1 or by the constitutively active YAP 5SA, we have now included an associated graph in the new Fig. 5e, detailing the statistically significant differences among the different treatments. This new quantification graph provides further evidence that cells over-expressing β -arr1 or YAP were more resistant to CIS or CIS+MAC compared with parental cells. Accordingly to the reviewer suggestion, we have amended the old Fig.5d-e to maximise the impact of the results, and we have focused on the more relevant and detailed new Fig. 5d-e.

Comment: Minor grammatical errors throughout the manuscript. Please edit for grammar.

Response: Thanks for pointing this out. The entire manuscript has been edited for grammatical errors.

Best regards,

Anna Bagnato